# Two-dimensional materials-based probabilistic synapses and reconfigurable neurons for measuring inference uncertainty using Bayesian neural networks

Amritanand Sebastian[1] ✉, Rahul Pendurthi[1], Azimkhan Kozhakhmetov[2], Nicholas Trainor[2,3], Joshua A. Robinson[2,4,5], Joan M. Redwing[2,3,6] & Saptarshi Das [1,2,6] ✉

Artificial neural networks have demonstrated superiority over traditional computing architectures in tasks such as pattern classification and learning. However, they do not measure uncertainty in predictions, and hence they can make wrong predictions with high confidence, which can be detrimental for many mission-critical applications. In contrast, Bayesian neural networks (BNNs) naturally include such uncertainty in their model, as the weights are represented by probability distributions (e.g. Gaussian distribution). Here we introduce three-terminal memtransistors based on two-dimensional (2D) materials, which can emulate both probabilistic synapses as well as reconfigurable neurons. The cycle-to-cycle variation in the programming of the 2D memtransistor is exploited to achieve Gaussian random number generator-based synapses, whereas 2D memtransistor based integrated circuits are used to obtain neurons with hyperbolic tangent and sigmoid activation functions. Finally, memtransistor-based synapses and neurons are combined in a crossbar array architecture to realize a BNN accelerator for a data classification task.

Machine learning has seen unprecedented growth and success in recent years owing to the development of artificial neural networks (ANNs). By mimicking the biological neural architecture and employing deep learning algorithms, ANNs have demonstrated notable advantages over standard computing methods for tasks such as image classification, facial recognition, data mining, weather forecasting, and stock market prediction[1–5]. While ANNs offer high performance, especially in terms of high prediction accuracy, they often suffer from overfitting due to a lack of generalization as they do not model uncertainty. Large datasets and various regularization techniques are often required to reduce overfitting in ANNs[6]. However, this can limit the use of ANN in applications where the data is scarce. In addition, uncertainty estimation is important in applications like autonomous driving, and medical diagnostics, where machine learning must be complemented with uncertainty-aware models or human intervention[7,8]. The integration of probabilistic computing paradigms with ANNs allows regularization and enables us to model uncertainty in predictions[9–12]. This is achieved in Bayesian neural networks (BNNs) by incorporating Bayes theorem to the traditional neural network scheme[12,13]. BNNs are capable of modeling uncertainty and avoiding overfitting, while working well with small

[1]Deparment of Engineering Science and Mechanics, Penn State University, University Park, PA 16802, USA. [2]Department of Materials Science and Engineering, Penn State University, University Park, PA 16802, USA. [3]2D Crystal Consortium Materials Innovation Platform, Penn State University, University Park, PA 16802, USA. [4]Department of Chemistry, Penn State University, University Park, PA, USA. [5]Department of Physics, Penn State University, University Park, PA, USA. [6]Department of Electrical Engineering and Computer Science, Penn State University, University Park, PA, USA. ✉e-mail: amritsebastian@gmail.com; sud70@psu.edu

datasets[14]. In fact, BNNs are extremely powerful as they represent an ensemble model, which is equivalent to the combinations of numerous ANNs, but with a small number of parameters. Unlike ANNs, where the synaptic weights are point estimates (single-valued), in BNNs, the weights ($W$) are represented by probability distributions, as shown in Fig. 1.

Over the years we have witnessed the development of neural network accelerators aimed at improving the size, energy consumption, and speed of neural networks, especially for edge computing applications[15–17]. Since the training process in neural networks is energy and resource intensive, these works typically rely on off-chip training and on-chip inference. Hence, BNN accelerators have also mostly focused on implementing Bayesian inference on-chip[18–23]. A crucial component of the BNN accelerator is an on-chip Gaussian random number generator (GRNG)-based synapse that can sample weights from a Gaussian distribution. In addition, a BNN requires a circuit to implement a neuron, i.e., to perform the multiply and accumulate (MAC) operation and the neural activation. BNN implementations based on Si complementary metal oxide semiconductor (CMOS) and field-programmable gate array (FPGA) typically require elaborate hardware for GRNGs, MAC operation, and the activation function, rendering them area and energy inefficient[18–20]. Moreover, these demonstrations are based on the von-Neumann architecture with separate memory and logic units, requiring frequent shuttling of data between the two. BNN accelerators based on emerging and non-von Neumann memristive and spintronic synapses utilize cycle-to-cycle variability in switching to generate Gaussian random numbers (GRNs)[21–23]. However, these GRNG-based synapses are limited to mean ($\mu$) of 0 and standard deviation ($\sigma$) of 1 and require extensive CMOS-based peripheral circuitry to obtain unrestricted $\mu$ and $\sigma$ values. For example, multiplication and addition operations are used to transform $N(0,1)$ to $N(\mu,\sigma) = \sigma*N(0,1) + \mu$. Finally, two-terminal memristors also lack the capability to emulate neurons for the activation functions. Therefore, energy and area efficient acceleration of BNNs will benefit from a standalone hardware platform, which can offer both neurosynaptic functionalities as well as programmable stochasticity.

In this work, we introduce three-terminal memtransistor technology based on two-dimensional (2D) monolayer $MoS_2$ and $WSe_2$ offering all computational primitives needed for a BNN accelerator. First, we realize an ultra-low power GRNG-based synapse by exploiting the cycle-to-cycle variability in programming/erasing operation in the $MoS_2$ memtransistor. Next, using a circuit comprising of two memtransistors we achieve reconfigurable $\mu$ and $\sigma$. Activation functions such as hyperbolic tangent (tanh) and sigmoid are also realized using the integration of $n$-type $MoS_2$ and V-doped $p$-type $WSe_2$ memtransistors. Finally, we demonstrate a crossbar array architecture in order to implement on-chip BNN inference. Furthermore, the entire network is simulated using LTSpice and uncertainty decomposition is performed to identify the various sources of uncertainty.

## Results

### 2D memtransistors

This 2D memtransistor has a back-gated geometry, where, $Al_2O_3$ is used as the gate dielectric and TiN/Pt is used as the gate electrode (see the "Methods" section and our earlier publications[24–27] for details on fabrication). We have used monolayer $MoS_2$[28,29] and V-doped $WSe_2$[30] grown using metal organic chemical vapor deposition (MOCVD) described in our previous reports as the channel materials. The choice of 2D materials is motivated by recent demonstrations highlighting the technological viability of 2D materials[28,31,32] and their wide-scale adoption in brain-inspired computing[33–42]. The transfer characteristics, i.e., drain current ($I_{DS}$) versus gate-to-source voltage ($V_{GS}$) for drain-to-source voltage ($V_{DS}$) of 1 V for 250 $MoS_2$ memtransistor demonstrating unipolar $n$-type behavior and the distribution of their threshold voltage ($V_{TH,n}$) are shown in Fig. 2a, b. The transfer characteristics for 20 V-doped $WSe_2$ memtransistors demonstrating unipolar $p$-type behavior is shown in Fig. 2c. Here the $WSe_2$ is doped with 1.1% V to realize unipolar p-type characteristics[30]. Figure 2d shows the distribution of the threshold voltage ($V_{TH,p}$) for $WSe_2$ memtransistors. Here, the threshold voltages are extracted using the constant-current method, at 0.1 nA $\mu m^{-1}$. The device-to-device variation is seen to be low in both cases.

The memtransistors offers analog and non-volatile charge-trap memory. The memtransistor can be programmed i.e., the threshold voltage of the device can be decreased by applying a program pulse to the back-gate with a large negative voltage ($V_P$). Figure 2e demonstrates the post-programmed transfer characteristics of a $MoS_2$ memtransistor for $V_{DS} = 0.1$ V, measured after programming with different $V_P$. Similarly, $MoS_2$ memtransistor can be erased i.e., the threshold voltage of the device can be increased by applying an erase pulse to the back-gate with a large positive voltage ($V_E$), as shown in Supplementary Fig. 1b. A pulse duration ($t$) of 100 $\mu s$ is used for both programming and erasing. The dependence of programming and erasing, on $t_P$ is shown in Supplementary Fig. 1c, d, respectively. The non-volatile nature of the $MoS_2$ memtransistor is shown in Fig. 2f, where the retention characteristics for five different conductance states are demonstrated for 2000 s. The characterization of memory of $WSe_2$ memtransistors is shown in Supplementary Fig. 2. The working principle of this analog and non-volatile memory has been described in detail in our earlier report[35].

### Gaussian random number generator-based synapse

We have demonstrated Gaussian synapses for a probabilistic neural network in our prior work, where the synapses were realized by mimicking the Gaussian function[33]. However, BNN accelerators require GRNGs and they typically rely on techniques such as cumulative density function inversion, central limit theorem (CLT)-based approximation, and the Wallace method to sample standard GRNs[18–20]. These methods typically require linear feedback shift registers, multipliers, and adders, involving numerous transistors to implement the GRNGs, rendering them area and energy inefficient. In contrast, here we use

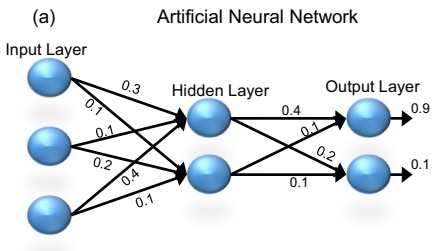

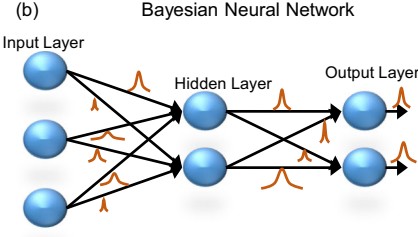

**Fig. 1 | Comparison of an artificial neural network (ANN) and a Bayesian neural network (BNN).** Schematic of **a** an ANN and **b** a BNN. The synapses of ANN are represented by single-valued weights while the synapses of a BNN is represented by probability distributions.

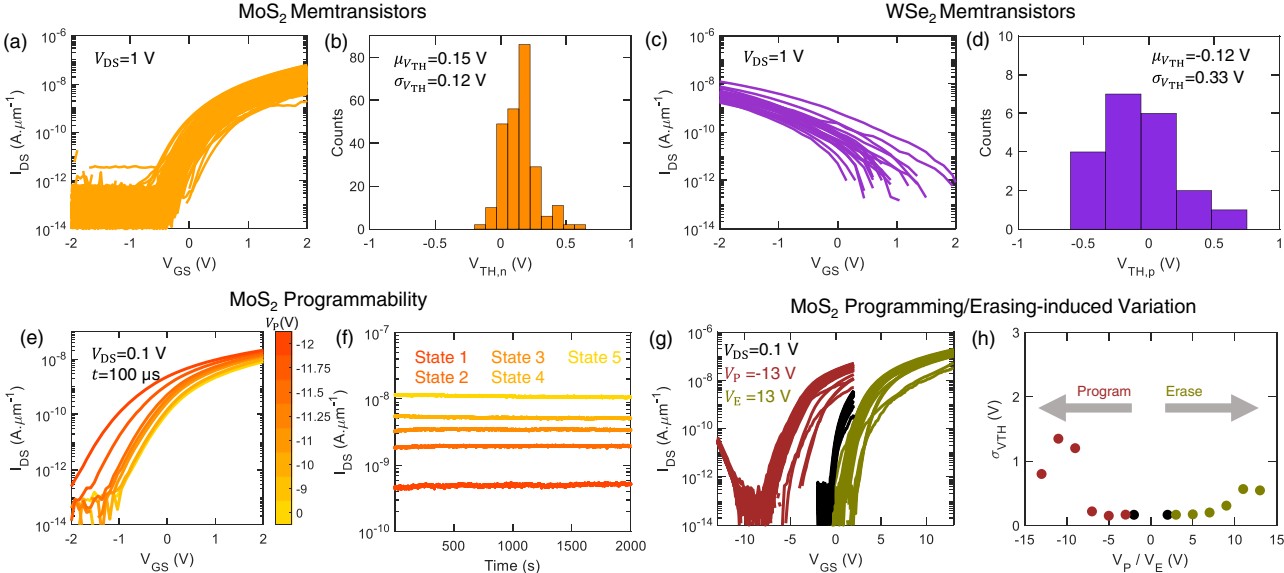

**Fig. 2 | Programmable memtransistors. a** Device-to-device variation in the transfer characteristics, i.e., drain current ($I_{DS}$) versus gate-to-source voltage ($V_{GS}$) for drain-to-source voltage ($V_{DS}$) of 1 V for 250 $MoS_2$ memtransistors, demonstrating unipolar $n$-type transport. Distribution of **b** threshold voltage ($V_{TH,n}$) extracted from 250 $MoS_2$ memtransistors. **c** Device-to-device variation in the transfer characteristics for 20 V-doped $WSe_2$ memtransistors, demonstrating unipolar $p$-type transport. Distribution of **d** threshold voltage ($V_{TH,p}$) extracted from 20 $WSe_2$ memtransistors. The corresponding means ($\mu$) and standard deviations ($\sigma$) are denoted in the inset. Memtransistors offers analog and non-volatile memory,

where their threshold voltage can be adjusted by applying a programming pulse to the back gate. **e** Transfer characteristics of a post-programmed $MoS_2$ memtransistor obtained by applying negative programming voltages ($V_P$) of different amplitudes for pulse duration ($t$) of 100 μs. **f** Analog retention characteristics, i.e., post-programmed $I_{DS}$ versus time measured at $V_{GS}$ = 0 V, for five different states. **g** The impact of programming/erasing on device-to-device variation, demonstrated using different $V_{GS}$ sweep ranges. **h** $\sigma_{V_{TH,n}}$ as a function of $V_P$ and erasing voltage ($V_E$) applied during $V_{GS}$ sweeps. An increase in variation is seen for high $V_P/V_E$.

cycle-to-cycle variation in the programmability of our $MoS_2$ memtransistor to generate GRNs. While cycle-to-cycle variation is undesirable for traditional computing, it can be exploited to reduce the design complexity of a BNN accelerator[21,22,43]. To demonstrate the effect of programming on variation, we use dynamic programming on 40 $MoS_2$ memtransistors, where we measure the transfer characteristics with different $V_{GS}$ sweep ranges. To evaluate the effect of $V_P$ ($V_E$), the maximum positive (negative) $V_{GS}$ is fixed at +2 V (−2 V), while the maximum negative (positive) $V_{GS}$ is stepped from −3 to −13 V (3 to 13 V). As shown in Fig. 2g, high $V_P$ and $V_E$ (±13 V) increases the device-to-device variation (post programming/erasing). High $V_P$ and $V_E$ (beyond ±7 V) results in significant $V_{TH}$ shift (see Supplementary Fig. 3), while also increasing $\sigma_{V_{TH}}$, as shown in Fig. 2h. This increase in device-to-device variation for high $V_P$ and $V_E$ is also accompanied by an increase in the cycle-to-cycle variation.

To utilize the cycle-to-cycle variation, the gate of a $MoS_2$ memtransistor is subjected to successive erase-program-read pulse cycles with $V_E$ = 13 V, $V_P$ = −13 V, and read voltage ($V_R$) of 0 V as shown in Fig. 3a. The corresponding $V_{DS}$ values were 0, 0 and 0.1 V, respectively. The conductance ($G$) of the memtransistor, measured at each read step, is shown in Fig. 3b for 200 cycles. As evident from the histogram shown in Fig. 3c, $G$ follows a Gaussian distribution, with mean, $\mu_G$ = 3.5 nS and, standard deviation, $\sigma_G$ = 0.9 nS. The quantile-quantile (Q-Q) plot of $G$ further confirms the Gaussian distribution. The quantiles of $G$ (represented using circles) are plotted against the theoretical quantiles from a Gaussian distribution as shown in Fig. 3d. As expected, it closely follows a straight line. Note that the slope of the Q-Q plot represents $\sigma_G$ and $G$ corresponding to quantile 0 represents $\mu_G$. Further characterization of 2D memtransitor-based GRNG has been done in our previous report[44]. Here, the random distributions are observed as an effect of random nature of charge trapping which is typically observed in charge-trap memory devices[45,46]. Hence, these memtransistors can also be replaced with standard three-terminal charge-trap flash memories such as TaN-$Al_2O_3$-$Si_3N_4$-$SiO_2$-Si (TANOS) and Si-$SiO_2$-$Si_3N_4$-$SiO_2$-Si

(SONOS)[47–49]. Nevertheless, $MoS_2$ memtransistors are used to generate a physical random variable that samples analog conductance values from a Gaussian distribution, i.e., $G \sim N(\mu_G, \sigma_G)$. Moreover, the $MoS_2$ memtransistor can be used as a synapse, which scales the input by its synaptic weight. If input is applied as voltage to the drain terminal of the memtransistor, the output current is scaled by $G$, i.e., $I_{DS} = G.V_{DS}$, as shown in Fig. 3a. Therefore, by combining the cycle-to-cycle variation in $G$ with the synaptic functionality of the memtransistor, we are able to realize a GRNG-based synapse. To implement a BNN accelerator, it's important to tune both $\mu_G$ and $\sigma_G$ of the GRNG-based synapse independently. $\mu_G$ and $\sigma_G$ can be tuned by modulating $V_P$ in the erase-program-read pulse cycle, as shown in Fig. 3e. However, $\mu_G$ and $\sigma_G$ are found to be coupled, and the coefficient of variation ($C_v = \mu_G/\sigma_G$) depends on $V_P$, as shown in Fig. 3f. A similar trend is seen in $\mu_G$, $\sigma_G$ and $C_v$ as a function of $V_E$, as shown in Supplementary Fig. 4.

Figure 4a shows the design of our GRNG-based synapse with independent control over its $\mu$ and $\sigma$, using two $MoS_2$ memtransistors, $T_+$ and $T_-$. While prior demonstrations rely on additional mathematical manipulations of the generated GRNs to establish control over their $\mu$ and $\sigma$, we are able to achieve it without any additional manipulations or circuitry[21–23,43]. It is common practice in neural network accelerators to use two devices per synapse in order to map both positive and negative weights[49]. Here, the input to the synapse, $V_{in}$ is applied as $+V_{in}$ and $-V_{in}$ to $T_+$ and $T_-$, respectively, as shown in Fig. 4a. The current at the output node ($I_{out}$) is then given by sum of currents through $T_+$ and $T_-$ i.e., $I_{T_+}$ and $I_{T_-}$, according to the Kirchhoff's current law (KCL) given by Eq. 1.

$$I_{out} = I_{T_+} + I_{T_-} = G_+ . V_{in} - G_- . V_{in} = (G_+ - G_-).V_{in} = G_{eff}.V_{in} \quad (1)$$

Here $G_+$ and $G_-$ are the conductance of $T_+$ and $T_-$ respectively and $G_{eff}$ is the effective conductance of the synapse. While the conductance of a device is always positive, by modulating $G_+$ and $G_-$,

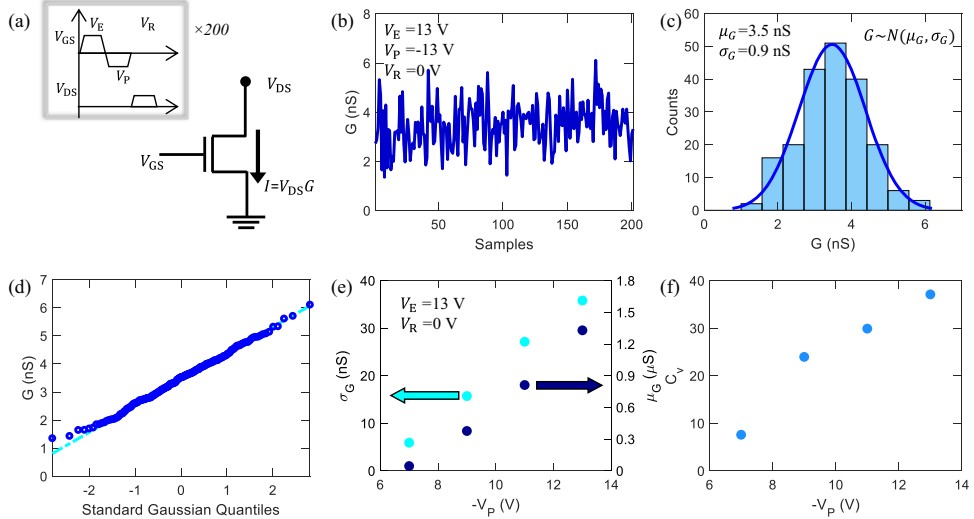

**Fig. 3 | Gaussian random number generator (GRNG) using MoS2 memtransistor. a** Schematic of a $MoS_2$ memtransistor used as a GRNG. To generate gaussian random numbers (GRNs), the gate of a $MoS_2$ memtransistor is subjected to an erase-program-read pulse cycle 200 times with $V_E$ of 13 V, $V_P$ of −13 V, and read voltage ($V_R$) of 0 V, while the drain is subjected to $V_{in}$ of 0, 0 and 0.1 V, respectively. **b** The corresponding conductance ($G$) of the $MoS_2$ memtransistor measured at each read step, demonstrating random fluctuations in $G$. **c** Histogram

demonstrating that $G$ follows a Gaussian distribution, with $\mu_G$ = 3.5 nS and $\sigma_G$ = 0.9 nS. **d** The quantile-quantile (Q-Q) plot of $G$, confirming its Gaussian distribution. The quantiles of $G$ (represented using circles) are plotted against the theoretical quantiles from a Gaussian distribution, which follows the straight line expected for a Gaussian distribution. **e** Dependence of $\mu_G$ and $\sigma_G$ and the corresponding **f** coefficient of variation ($C_v$) on $V_P$. Here, $V_P$ is changed in the erase-program-read pulse cycle. $\mu_G$ and $\sigma_G$ are seen to be coupled.

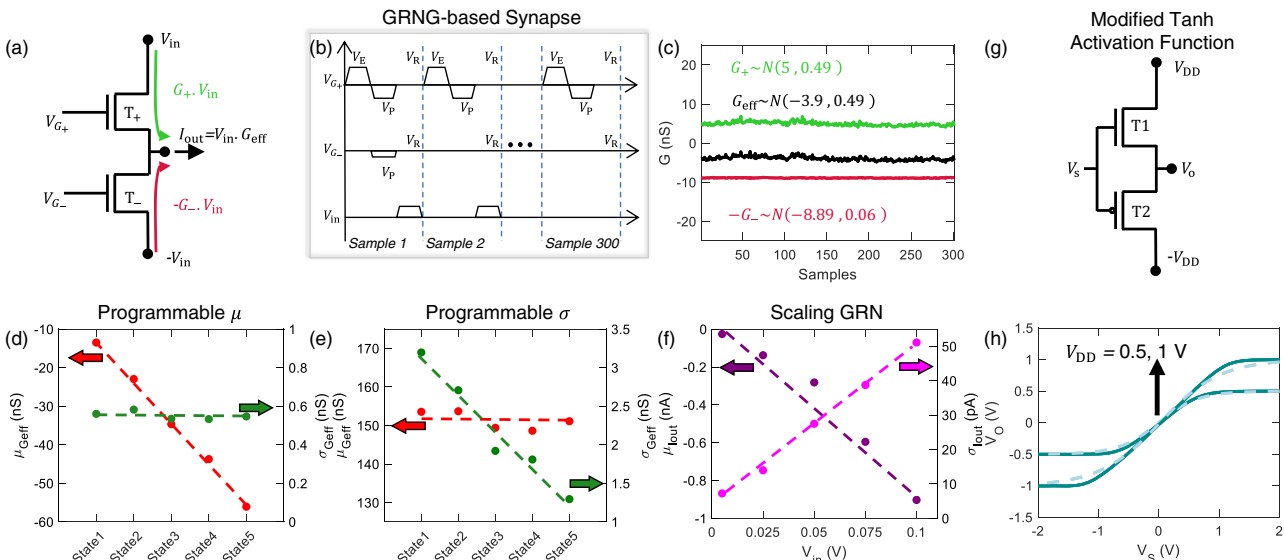

**Fig. 4 | GRNG-based synapse and modified tanh activation function. a** Schematic of the GRNG-based synapse. The input to the synapse, $V_{in}$ is applied as $+V_{in}$ and $-V_{in}$ to the memtransistors, $T_+$ and $T_-$ with conductance $G_+$ and $G_-$ (modulated using $V_{G_+}$ and $V_{G_-}$), respectively. The effective conductance of this synapse is given by $G_{eff} = G_+ - G_-$, allowing positive and negative conductance. **b** Waveform applied to the synapse to generate GRNs with independent control over $\mu$ and $\sigma$. **c** $G_+$, $-G_-$ and $G_{eff}$ of the synapse sampled 300 times, where $T_+$ controls $\sigma$ and $T_-$ controls $\mu$. GRNG-based synapse showing **d** independent control of $\mu_{G_{eff}}$ for constant $\sigma_{G_{eff}}$ and

**e** independent control of $\sigma_{G_{eff}}$ for constant $\mu_{G_{eff}}$. **f** Linear scaling of the synaptic output ($I_{out}$) distribution as the function of $V_{in}$. **g** Schematic of circuit for the modified tanh activation function using a $n$-type $MoS_2$ memtransistor (T1) and a V-doped $p$-type $WSe_2$ memtransistor (T2), where the input voltage ($V_S$) is applied to the gate terminal of T1 and T2. **h** The transfer characteristics of the circuit (solid line) i.e., output voltage ($V_O$) versus $V_S$, closely models the tanh activation function (dotted line).

using $V_{G_+}$ and $V_{G_-}$ (by applying different $V_P$), we can obtain both positive and negative $G_{eff}$. To control $\mu_{G_{eff}}$ and $\sigma_{G_{eff}}$, $T_+$ is subjected to successive erase-program-read pulse cycles, while $T_-$ is programmed to a given state and subsequently only read, using the waveforms shown in Fig. 4b. This results in $G_+$ being drawn from a Gaussian distribution, with $\mu_{G_+}$ = 5 nS and $\sigma_{G_+}$ = 0.49 nS i.e., $G_+ \sim N(5,0.49)$ nS and $G_-$ having a constant value of ≈ 8.89 nS, as shown in Fig. 4c. $G_{eff}$ is expected to be drawn from a distribution with $\sigma_{G_{eff}} = \sigma_{G_+}$ and

$\mu_{G_{eff}} = \mu_{G_+} - G_-$. This is confirmed by our measurements as shown in Fig. 4c, $G_{eff} \sim N(-3.9,0.49)$ nS. Note that, $G_-$ is not perfectly constant due to the presence of random telegraph fluctuations. However, the fluctuations were found to have a standard deviation of 0.06 nS, making its contribution negligible. The histograms and Q-Q plots of $G_+, G_-$, and $G_{eff}$ are shown in Supplementary Fig. 5. Figure 4d shows the independent control of $\mu_{G_{eff}}$ for constant $\sigma_{G_{eff}}$ using the GRNG-based synapse. Here, $T_+$ is subjected to the same erase-program-read

cycle, to obtain constant $\sigma_{G_{eff}}$, whereas $T_-$ is programmed to different states (using $V_P$) to tune $\mu_{G_{eff}}$. Figure 4e shows the independent control of $\sigma_{G_{eff}}$ for constant $\mu_{G_{eff}}$. In order to modulate $\sigma_{G_{eff}}$, $\sigma_{G_+}$ is changed by applying different erase-program-read cycles (different $V_P$) to $T_+$. Since this leads to an unfavorable change in $\mu_{G_+}$, $T_-$ is reprogrammed to account for the change in $\mu_{G_+}$, to maintain a constant $\mu_{G_{eff}}$. In a synapse, the distribution of $I_{out}$ is expected to scale linearly with $V_{in}$, as given by Eq. 2.

$$I_{out} = G_{eff}.V_{in} = N(\mu_{G_{eff}}, \sigma_{G_{eff}}).V_{in} = N(\mu_{G_{eff}}.V_{in}, \sigma_{G_{eff}}.V_{in}) = N(\mu_{I_{out}}, \sigma_{I_{out}}) \tag{2}$$

This is demonstrated in Fig. 4f, where $\mu_{I_{out}}$ and $\sigma_{I_{out}}$ show linear dependence with respect to $V_{in}$. The output characteristics of a MoS$_2$ memtransistor is shown in Supplementary Fig. 6 for positive and negative $V_{DS}$. While the current is highly non-linear and asymmetric for large $\pm V_{DS}$ values, it is seen to be sufficiently linear and symmetric between $\pm 0.1$ V. Hence, we limit the maximum $V_{in}$ to 0.1 V. The low $V_{in}$ allows us to operate the synapse with extremely low currents, as shown in Fig. 4f, offering significant energy efficiency. Overall, we demonstrate independent control over $\mu_{G_{eff}}$ and $\sigma_{G_{eff}}$ to implement a GRNG-based synapse with just two MoS$_2$ memtransistors resulting in significant area and energy efficiency. Supplementary Fig. 7 demonstrates the stability of analog random number generation for numerous analog states for a total of 20,000 cycles. In addition, it demonstrates that both $\mu$ and $\sigma$ increase as a function of $V_{DS}$ and $|V_P|$, retaining the synaptic behavior for 20,000 cycles. Also, as cycle-to-cycle variation is used for random number generation, Supplementary Fig. 7 also represents that MoS$_2$ memtransistor has an endurance of at least 20,000 cycles. Higher endurance is desired for BNNs and this can be achieved through further optimization of MoS$_2$ memtransistors.

## Neurons with modified hyperbolic tangent activation function

The hardware for activation function in neural accelerators is generally realized using standard CMOS-based analog and digital components, and hence these implementations do not utilize the advantages offered by emerging materials[49]. Moreover, hyperbolic tangent (tanh) and sigmoid functions are highly non-linear, significantly complicating their hardware demonstration[50]. We demonstrate a circuit for a modified tanh (m-tanh) activation function using a $n$-type MoS$_2$ memtransistor (T1) and a V-doped $p$-type WSe$_2$ memtransistor (T2) as shown in Fig. 4g. The transfer function of the circuit i.e., output voltage ($V_O$) versus input voltage ($V_S$) closely follows the tanh activation function as shown in Fig. 4h. The maximum of the m-tanh activation function is determined by the drain voltage ($V_{DD}$). Here, $V_S$ is applied to the gate of T1 and T2. T1 and T2 are programmed to ensure that the m-tanh function passes through the origin. Note that when $V_S = -2$ V, T1 operates in the off-state and T2 operates in the on-state, resulting in $V_O = -V_{DD}$, whereas for $V_S = 2$ V, T1 operates in the on-state and becomes more conductive than T2, which results in $V_O = V_{DD}$. While this output characteristics is well-known[49,51], to the best of our knowledge it has not been used to implement activation functions, as the activation functions are typically implemented using look-up-tables[49]. In addition, modified sigmoid activation function can be realized by applying 0 V to the drain terminal of T1, as shown in Supplementary Fig. 8.

## Crossbar array architecture

The crossbar array architecture is routinely used in neural network accelerators to perform the MAC operation of a neuron. Figure 5a shows the circuit used to implement a portion of a BNN shown in Fig. 5b, where $M = 4$ input neurons are connected to $N = 1$ output neuron. Each input neuron is multiplied with their corresponding synaptic weight distributions and the resultants are summed at the output neuron (MAC operation). The resultant of the MAC operation is passed through the m-tanh activation function, to obtain the output.

To implement this on circuit, as shown in Fig. 5a, the conductance distribution of a synapse in the $i^{th}$ row, $j^{th}$ column and $k^{th}$ layer ($G_{ij}^{(k)}$), given by the combination of $G_{ij_+}^{(k)}$ and $G_{ij_-}^{(k)}$ is modulated using $V_{Gj_+}^{(k)}$ and $V_{Gj_-}^{(k)}$ lines. Inputs to $i^{th}$ row and $k^{th}$ layer are applied as voltages ($\pm V_i^{(k)}$). The current through the $j^{th}$ column, due to these synapses is then given by the dot product of $V_i^{(k)}$ and $G_{ij}^{(k)}$, according to KCL. To obtain a voltage proportional to this dot product, we use a sense transistor, as shown in Fig. 5a. The voltage-drop ($V_{Sj}^{(k)}$) across this sense transistor is given by Eq. 3.

$$V_{Sj}^{(k)} = \frac{\sum_{i=1}^{M} V_i^{(k)} G_{ij}^{(k)}}{G_{Sj}^{(k)} + \sum_{i=1}^{M} G_{ij}^{(k)}} \tag{3}$$

Here, $G_{Sj}^{(k)}$ is the conductance of the sense transistor, and is modulated using $V_{sense}^{(k)}$. Using $V_{Sj}^{(k)}$ allows us to seamlessly integrate the circuit for m-tanh activation function into the crossbar array as shown in Fig. 5a, to obtain the corresponding output ($V_{Oj}^{(k)}$). There are some non-idealities which are also accounted for. First, the synaptic weight distribution ($W_{ij}^{(k)}$) is mapped to the crossbar array by using a conductance scaling factor ($\alpha$) to obtained $G_{ij}^{(k)}$. Second, the denominator of $V_{Sj}^{(k)}$ (Eq. 3) presents a non-ideality, which can be expressed as the product of $\alpha$ and a non-ideality factor ($\gamma^{(k)}$). By mapping the input ($x_i^{(k)}$) to $V_i^{(k)}$, using $\gamma^{(k)}$ as the scaling factor, ideal $V_{Sj}^{(k)}$ and $V_{Oj}^{(k)}$ can be obtained as shown in Eq. 4.

$$V_{Sj}^{(k)} = \frac{\sum_{i=1}^{M} V_i^{(k)} G_{ij}^{(k)}}{G_{Sj}^{(k)} + \sum_{i=1}^{M} G_{ij}^{(k)}} = \frac{\sum_{i=1}^{M} \left(x_i^{(k)} \gamma^{(k)}\right) \cdot \left(W_{ij}^{(k)} \cdot \alpha\right)}{\gamma^{(k)} \cdot \alpha} = \sum_{i=1}^{M} x_i^{(k)} W_{ij}^{(k)} \tag{4a}$$

$$V_{Oj}^{(k)} = \tanh\left(V_{Sj}^{(k)}\right) = \tanh\left(\sum_{i=1}^{M} x_i^{(k)} W_{ij}^{(k)}\right) \tag{4b}$$

$G_{Sj}^{(k)}$ is used to make sure that each column of the crossbar array has the same $\gamma^{(k)}$. With this proposed scheme, we can evaluate the dot product between $x_i^{(k)}$ and $W_{ij}^{(k)}$ in the voltage domain and use the m-tanh activation function to obtain the ideal output, $V_{Oj}^{(k)}$. Note that this scheme is not limited to the implementation of a BNN and can be adopted to implement standard ANN crossbar arrays with tanh and sigmoid activation functions.

## Neural network evaluation

We evaluate the performance of our BNN implementation using the PIMA Indian Diabetes dataset. This dataset consists of nine parameters such as number of pregnancies, glucose levels, insulin levels, body mass index, age, etc. To classify this dataset, we use a fully connected $8 \times 10 \times 2$ BNN, i.e., it has an input layer with 8 neurons, one hidden layer with 10 neurons, and an output layer with 2 neurons. The dataset with 767 instances is divided into 720 for training and 47 for testing. "Bayes by Backprop" algorithm, with a Gaussian prior is used to train the synaptic weight distributions [12,52] (see the "Methods" for details). The BNN is trained off-chip for 300 epochs as shown in Fig. 5c, to obtain train accuracy of 75.41% and test accuracy of 80.85%. Similar accuracy numbers have been reported in prior works[53–55].

Figure 5d shows the circuit of the BNN used to classify the PIMA Indians dataset. Here, the synapses are arranged in the crossbar array architecture. The BNN circuit to classify PIMA Indian diabetes dataset is evaluated using LTSpice simulations (See "Methods" for details). Note that, at the output layer we do not use the tanh activation function. Instead, following Eq. 6, the output of the BNN, $V_{Sj}^{(k)}$ is sampled $Z = 100$ times to obtain a distribution, and its mean is used to make the classification. Using the LTSpice simulations, we are able replicate the

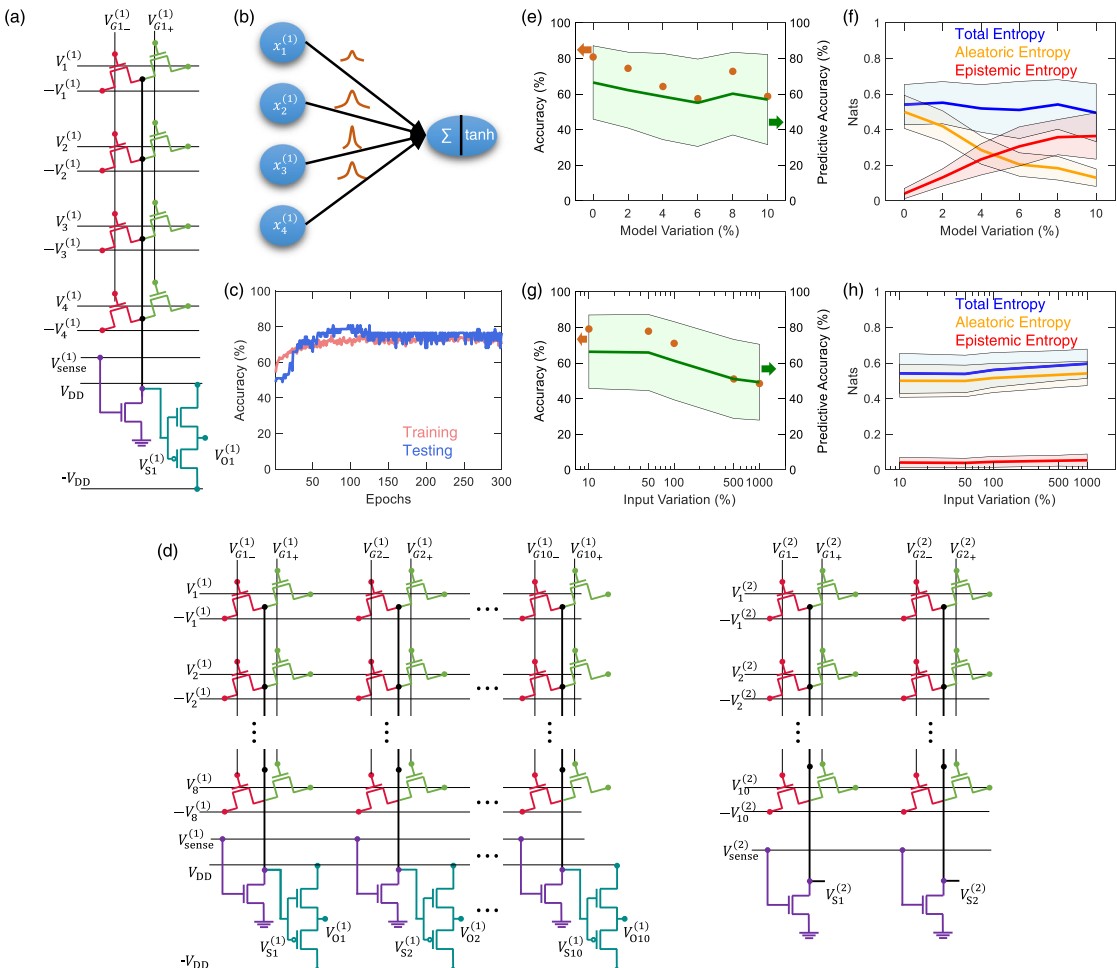

**Fig. 5 | Crossbar array architecture to implement the BNN. a** Schematic of a portion of a BNN, where multiple input neurons are connected to one output neuron through multiple synapses. **b** The corresponding circuit implementation using the crossbar array architecture. Here, a sense transistor is used to obtain a voltage ($V_{Sj}^{(k)}$) proportional to the dot product between the inputs ($V_i^{(k)}$) and synaptic distributions ($G_{ij}^{(k)}$), where $i$, $j$, and $k$ represents the row, column, and layer, respectively. $V_{Sj}^{(k)}$ is applied to the tanh activation circuit to obtain the output ($V_{Oj}^{(k)}$). **c** Training and testing curves for 300 epochs of the BNN constructed to classify the PIMA Indians dataset. **d** Circuit implementation of the BNN to perform inference on-chip. **e** Accuracy and predicative accuracy as function of model variation. Here, the effect of variation in synaptic devices, sense resistors, and activation function is demonstrated. **f** Total entropy, aleatoric entropy and epistemic entropy as a function of model variation. **g** Accuracy and predictive accuracy as a function of input variation. **h** Total entropy, aleatoric entropy and epistemic entropy as a function of input variation.

test accuracy of 80.85%. The BNN circuit consumes a miniscule 18.37 nJ/test sample in order to make the classification (see "Methods" for details).

While the cycle-to-cycle variation in memtransistors is useful for the generation of Gaussian random numbers, the device-to-device variation observed along with it is undesirable for the operation of a neural network. Supplementary Note 1 and Supplementary Fig. 9 discusses a method to model and reduce the device-to-device variation. Since, some variation is still expected to exist, we evaluate its effect on the performance of the circuit for the Bayesian neural network. We implement up to 10 % variation for the parameters of the synapse: $\mu_{G_+}$, $\sigma_{G_+}$, and $G_-$, which results in a corresponding variation in $G_{eff}$, the parameters of the tanh function: $V_{TH,n}$ and $V_{TH,p}$, and for the $n$-type FET and the $p$-type FET, and the conductance of the sense resistor $G_S$. These parameters are drawn from Gaussian distributions where the mean is given by expected parameter value and standard deviation of up to 10% is considered. Figure 5e shows the effect of device-to-device (model) variation on the test accuracy and predictive accuracy. The predictive accuracy demonstrates how well the predictions of the expected correct classes are made. Here, the BNN is simulated and averaged over 5 runs. While, we observe a decrease in the test

accuracy, it is not seen to significantly impact the operation of the BNN and an accuracy of ≈60% is maintained for 10% variation.

In a BNN, we can use entropy estimation and entropy decomposition to quantify uncertainty and to find its source, respectively (see the "Methods" section for details). The distribution of $V_{Sj}^{(k)}$ at the output layer can be used to calculate the uncertainty in classification[56,57]. Figure 5f shows the total entropy, aleatoric entropy, and epistemic entropy as a function of model variation, calculated using Eqs. 8 and 9. In model variation, it was determined that the variation in synapses is more determinantal to the performance of BNN compared to variations in the neuron, due to the binary nature of our classification problem. For a multiclass problem, variations in the neuron are expected to have a larger impact on the accuracy. In Fig. 5f, we would expect the aleatoric uncertainty to remain unchanged and total entropy to increase. However, their extraction is impacted by the increased model variation. Nevertheless, as expected, the epistemic entropy increases as the model variation increases. As shown in Fig. 5g, h, an increase in the input variation results in the degradation of accuracy, along with an increase in the total and aleatoric entropy, while a constant epistemic entropy is maintained, as expected. Hence, in addition to the estimation of total entropy

using a BNN, with uncertainty decomposition various sources of entropy can be identified.

## Discussion

This work demonstrates the development of computational primitives needed for a BNN accelerator, using 2D memtransistors. The cycle-to-cycle variation in the programming of the $MoS_2$ memtransistor is exploited as a source of randomness and a circuit comprising of two such memtransistors is used to obtain an ultra-low-power and stochastic synapse, which allows sampling of both positive and negative weights from a Gaussian distribution with reconfigurable mean and standard deviation. We also developed circuits to implement the modified hyperbolic tangent and sigmoid activation functions based on the integration of $MoS_2$ and $WSe_2$ memtransistors. In addition, we integrate these components into a crossbar array architecture to perform efficient MAC operations and we develop a BNN circuit to perform on-chip inference to classify the PIMA Indians dataset and evaluate its performance using circuit simulations. Finally, we have also performed uncertainty decomposition in order to identify the various sources of uncertainty.

## Methods

### Device fabrication

$MoS_2$ memtransistors are fabricated using photolithography and e-beam lithography. Photo-lithography is used to define the back-gate islands. A $p^{++}$ Si substrate is first spin coated with LOR 5 A and baked at 180 °C for 120 s, and subsequently spin coated with SPR 3012 and baked at 95 °C for 60 s. Using Heidelberg MLA 150, the desired regions are exposed to 405 nm light. The exposed regions are developed using 1:1 CD 26 and DI water. To form the back gate islands, 20 nm TiN followed by 50 nm Pt is deposited through sputtering. 50 nm $Al_2O_3$ gate dielectric is deposited using atomic layer deposition. $Al_2O_3$ is etched from back gate contact regions using $BCl_3$ etch, where the etch region was defined by photolithography. Following this MOCVD $MoS_2$ is transferred onto this substrate and the $MoS_2$ transistors are fabricated as discussed in our previous reports[28,51]. The fabrication of $WSe_2$ memtransistors have been discussed in our previous report[30].

### Electrical characterization

Lake Shore CRX-VF probe station and Keysight B1500A parameter analyzer were used to perform the electrical characterization at room temperature. The device-to-device variation measurements were performed using the FormFactor Cascade Summit 12000 semi-automated probe station.

### BNN training and inference

According to the Bayes theorem, the weights of a BNN are given by posterior probability distributions given by Eq. 5.

$$p(W|D) = \frac{p(D|W) \cdot p(W)}{p(D)} \tag{5}$$

Here $D$ is the training data, $p(W|D)$ is the posterior distribution, $p(D|W)$ is the likelihood, $p(W)$ is the prior, and $p(D)$ is the evidence. The true posterior distribution is untraceable in BNNs and hence, methods such as variational inference[12] and Markov chain Monte Carlo (MCMC) sampling[58] are used to approximate the posterior distribution. Variational inference is typically preferred due to better convergence and scalability compared to MCMC[20]. In the variational inference method, $p(D|W)$ is estimated using a family of variational posterior distributions (typically Gaussian distributions), $q(W;\theta)$, where $\theta$ represents the variation parameters. For a Gaussian distribution, the variation parameters are its mean and standard deviation, i.e., $\theta = \mu, \sigma$. The estimation is performed by minimizing the Kullback-Leibler divergence between $p(D|W)$ and $q(W;\theta)$. In the training phase, for each synapse, $\mu$ and $\sigma$ are

learned using the traditional backpropagation method[12]. Here, $\sigma$ represents the uncertainty introduced by each synapse. To perform inference using a BNN, multiple forward passes of the trained network is evaluated. During each forward pass, each of the Gaussian weight distributions are sampled once. The output of the network or the predictive distribution is obtained by averaging the outputs of these forward passes obtained by sampling the weight distributions. It can be approximated by drawing $Z$ Monte Carlo samples and finding its mean given by Eq. 6.

$$p(y^*|x^*,D) \approx \frac{1}{Z}\sum_{z=1}^{Z} p(y^*|x^*,W^z) \tag{6}$$

Here, $x^*$ and $y^*$ are the test input and output, respectively, D is the training data, and $W^z$ represents the $z^{th}$ Monte Carlo weight sample.

### LTSpice Circuit Implementation

The synapses consisting of two memtransistors, $T_+$ and $T_-$ are modeled using resistors. During the read step, $V_R$ or of 0 V is applied to prevent read-disturb and $V_{in}$ is applied such that the transistors are operating in the linear part of the triode region. Hence, these memtransistors are modeled using resistors. Here, $T_+$ is modeled as a resistor which draws from Gaussian conductance distribution given by $G_+ \sim N(\mu_{G_+}, \sigma_{G_+})$ and $T_-$ is modeled as a resistor set to a constant conductance of $G_-$. The weight distributions are mapped to conductance distributions with $\alpha = 10^{-9}$. For the actual circuit implementation using memtransistors, in order to program and reprogram the crossbar array, we propose to use a scheme where all devices are erased with a high $V_E$ of 13 V. Following that, each device must be programmed with the separate $V_P$ values to set them to their expected states, requiring additional storge to store $V_P$ values. The sense transistor is modeled using a resistor, ensuring that each column of the crossbar array has the same $\gamma^{(k)}$. $\gamma^{(k)}$ is determined for each layer and multiplied with $x_i^{(k)}$ to obtain $V_i^{(k)}$. The tanh activation function is implemented using the combination of an $n$-type FET and a $p$-type FET. $V_{DD}$ of 1 V is used for the m-tanh activation circuit.

### Energy analysis

The total energy consumption of the BNN accelerator is given by Eq. 7.

$$E_{Total} = E_{syn} + E_{sense} + E_{tanh} + E_P + E_E$$

$$= \sum_N \left[ (I_{T_+} + I_{T_-})V_{in}t \right]_{syn} + [I_S V_S t]_{sense} + \left[ (I_{T1} + I_{T2})V_{DD}t \right]_{tanh} + E_P + E_E \tag{7a}$$

$$E_P + E_E = \sum_{N-100} t(I_P V_P + I_E V_E) + \sum_{100} \left[ Qt(I_P V_P + I_E V_E) \right]_{P,E} \tag{7b}$$

Here, $E_{Total}$ represents the total energy consumption per test sample. $E_{syn}$, $E_{sense}$, and $E_{tanh}$ represents the total energy consumption by the synapses, sense resistors, and the tanh circuit, respectively, during the inference step. $I_S$ represents the current through the sense resistor and $N$ is the total number of devices. $E_P$ and $E_E$ represents the energy consumption for programming and erasing operation on these components. $I_P$ and $I_E$ respectively, are the gate currents associated with programming and erase operations. Here, only half of the synaptic devices (100 devices) needs to be erased and programmed for $Z = 100$ Monte Carlo samples. The rest of the synaptic devices, sense transistors, and tanh circuit components are erased and programmed once and subsequently only used for inference. For $E_P$ and $E_E$ evaluations, maximum $I_P, V_P, I_E, V_E$ values expected are used. Using Eq. 7, $E_{syn}$ of 18 nJ, $E_{sense}$ of 0.025 nJ, and $E_{tanh}$ of 0.012 nJ are obtained from LTSpice simulations and $E_P + E_E$ of 0.34 nJ is estimated for a total of $Z = 100$

Monte Carlo samples, resulting in an $E_{Total}$ of 18.37 nJ. Note that while the program/erase voltages are high compared to other emerging memory technologies such as memristors and phase-change memory, the energy consumption associated with memtransistors is low as the program (gate) currents are on the order of pA and since the transistors are biased with a drain voltage of 0 V during programming/erasing, there is no drain current (See Supplementary Table 1 for energy analysis of various emerging memories)[59].

## Entropy calculation and decomposition

The softmax of predictive distribution can be used to calculate the uncertainty in classification or entropy given by Eq. 8.[56,57].

$$H(p(\hat{y}^*|x^*,D)) = -\sum_{j=1}^{J} p_j(\hat{y}^*|x^*,D)^* \log(p_j(\hat{y}^*|x^*,D)) \qquad (8)$$

Here, $\hat{y}^*$ is the softmax output and $J$ is the number of output classes. The entropy can be decomposed into epistemic entropy, i.e., uncertainty in model and aleatoric entropy, i.e., uncertainty in data as shown in Eq. 9.

$$H(p(\hat{y}^*|x^*,D)) = \prod(p(\hat{y}^*|x^*,D)) + E_{W \sim q(W;\theta)}\left[H(p(\hat{y}^*|x^*,W))\right] \qquad (9)$$

Here, on the right-hand side, the first term represents epistemic entropy and the second term represents aleatoric entropy. Aleatoric entropy is the average entropy for fixed weights and hence the uncertainty arises from the data. Epistemic entropy can be obtained by subtracting aleatoric entropy from total entropy, following Eq. 9.

## Data availability

The datasets generated during and/or analyzed during the current study are available from the corresponding authors on reasonable request.

## Code availability

The codes used for plotting the data are available from the corresponding authors on reasonable request.

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

## Acknowledgements

The work was supported by Army Research Office (ARO) through Contract Number W911NF1920338. Authors also acknowledge Mr. Shiva Subbulakshmi Radhakrishnan for help with m-tanh measurements. Authors also acknowledge the materials support from the National Science Foundation (NSF) through the Pennsylvania State University 2D Crystal Consortium–Materials Innovation Platform (2DCCMIP) under NSF cooperative agreement DMR-2039351. A.K. and J.A.R. acknowledge funding support from Intel, Inc. through the Semiconductor Research Corporation Task 2746.

## Author contributions

A.S. conceived the idea and designed the experiments. A.S. performed the experiments, analyzed the data. R.P. performed the fabrication and characterization of WSe$_2$ memtransistors. A.K. and N.T. synthesized the 2D materials under the supervision of J.A.R. and J.M.R., respectively. A.S. and S.D. discussed the results, agreed on their implications. All authors contributed to the preparation of the manuscript.

## Competing interests

The authors declare no competing interests
