## [Peer Review File · Nature Communications]

REVIEWER COMMENTS

Reviewer #1 (Remarks to the Author):

This paper describes the use of a previously demonstrated MoS₂ nonvolatile transistor for use in neural network processing circuits. The paper implements a Bayesian Neural Net using MoS₂ neurons and synapses. The BNN is used to explore uncertainty quantification. The accuracy is assessed using LTSpice on the IRIS dataset and with certain configurations is found to be reasonably high.

The topic of novel devices to implement BNNs is of interest to the community, as is the quantification of uncertainty in neural inference. This paper is very well written and well organized. However, before considering for publication in Nature Communications, there are some key issues which should be considered to clarify the novelty and significance of the results.

1. The authors spend a significant amount of time comparing the devices to two terminal memristors, and listing the advantages in that respect. However, the MoS₂ transistor as described is a three terminal charge trapping memory, and hence not particularly different from a standard three terminal charge trapping memory. While it is impressive that the authors have developed a CTM devices that incorporates MoS₂, it appears the circuits would all be possible to implement in a standard, commercial CTM memory, such as TANOS, SONOS, etc. The random distributions also seem likely an effect of charge trapping, which would be possible in a standard CTM. Please elaborate on whether this is the case, or there is something significantly different about the physics of the MoS₂ that cannot be replicated in a standard CTM memory.

2. The "tanh circuit" created by using T1 as a resistive load is a common drain amplifier, which is a standard transistor amplifier circuit with a well-known transfer function. I would recommend acknowledging this because it seems a bit like this is being presented as a novel circuit (which would seem odd to the microelectronic circuits community). I have also seen similar common drain circuit implemented with floating gate devices, giving similar VTC behavior, such as in Fig 5 of the referenced paper below.

3. For the LTSpice simulations, can you elaborate on the tanh function implementation; is this just modeled with a single resistor and NMOS? If the VTC looks reasonable (i.e. Fig 4h) then why is this a major source of inaccuracy? If this circuit is so inaccurate, why would it be used rather than the referenced previously used NMOS/PMOS version in ref 39?

4. A minor, optional note on the amplifier circuits, the more common convention is to have the high voltage on the top of the circuit such that current flows down. (It is opposite of this in Fig 4(g) and Supplemental Fig 7(a).

5. How was device to device variation modeled in LTSpice? Is there an experimental link between physical nonidealities and the modeling factors should be presented? Also, do the authors know the programming error of the devices, i.e. the delta between the expected and actual conductance after programming?

6. What is the gate voltage (or gate voltage range) of the synapses during read? What operating region is the transistor in at this gate voltage? It sounds like these are modulated, but in LTSpice, if you are treating the two device pair as a resistor, I am assuming the gate to source and drain to source voltages are such that both devices are operating in the linear part of the triode region, such that a resistor is a suitable element to represent the pair. However this is unclear from the description.

7. The authors should model something larger than the IRIS dataset, which is quite far from a dataset and network that would be used in a real world application. It has been observed that have seen that excellent accuracy on small datasets does not translate to high accuracy on larger datasets. There is little if any practical application for using a custom, efficient accelerator to process a dataset as small as IRIS. I would understand the very small dataset if this was an experimental demonstration, but given that it is a simulation only, the authors should include something larger.

8. Although the title conveys an interesting prospect, it does not seem well connected to the content and I would suggest modifying it. The work aims to quantify uncertainty, but does not "avoid inference inaccuracy".

9. Some of the figure captions are far too small to read. The labels on Fig 5d are equivalent to about a 4pt font.

Reference:

S. Shah, H. Toreyin, J. Hasler, and A. Natarajan, "Models and Techniques for Temperature Robust Systems on a Reconfigurable Platform," *Journal of Low Power Electronics and Applications*, vol. 7, no. 3, p. 21, Aug. 2017, doi: 10.3390/jlpea7030021.

Reviewer #2 (Remarks to the Author):

In this work, Sebastian and Das introduce a MoS₂-based device that can provide synaptic and neuronal functions to implement hardware Bayesian neural networks. These uses are illustrated in a simulation of a crossbar circuit. The device in itself is similar to previous works by the same group, but it is used in a new manner. The idea of exploiting stochastic effect inherent to device physics to implement probabilistic AI, as is done here, is a very strong concept, which has started to develop in recent literature. The device introduced by the authors has some nice features. However, I have concerns about the manuscript and some questions about the meaning of the work.

On a general note, I actually do not understand the benefits of using 2-D materials in this project. The authors explain: "The choice of MoS₂ as the element of memtransistor is motivated by recent demonstrations highlighting the technological viability of 2D materials [27-29] and their wide scale adoption in brain-inspired computing [30-34]."

This is too vague, in my opinion. In fact, the adoption of 2-D materials in brain-inspired computing is not that wide-scale: Refs 30-34 all have S. Das as senior author, if I am not mistaken.

The title of the article, "A Neural Network Accelerator to Avoid Inference Inaccuracy", does not seem related to the paper results. The authors do not present a neural network accelerator, and the paper never really talks about avoiding inference inaccuracy. This was a little surprising.

The authors should position this new work more clearly with regards to their paper "Gaussian synapses for probabilistic neural networks" (Ref 30), which has some similar keywords and ideas (but is different).

The programming voltages are very high, similar to FLASH memory, and much higher than memristors. The authors should benchmark their device with alternative approaches (FLASH, non-2D memristors, phase change memory...).

I have concerns about the proposed crossbar architecture. I understand that for each presented input, the crossbar devices need to be reprogrammed multiple times to provide a distribution at the

output (once per sample). Having to reprogram the crossbar numerous times to perform a single inference seems an enormous energy cost. Due to this concern, the paper should include an energy analysis with some benchmarks, in my opinion.

These multiple programming operations also raise the issue of device endurance, which should be discussed.

Also, if the synapses need to be reprogrammed each sample, we need additional memory arrays storing the mean value and standard deviations of each weight. This is an important cost, and this would also limit the energy efficiency of the authors' approach, as important data movement will be involved.

A significant concern is that Fig 5 shows no Bayesian result: we only see the final accuracy, but no distributions.

Also, why are Bayesian neural networks useful for this example? I think that conventional neural networks get excellent accuracy on this task.

The degradation in accuracy between software and LTSpice simulation is quite severe, even for a very simple task, due to the neuron behavior. This is an important limitation. Can this problem be fixed?

Fig 5g with 0% variation shows a test accuracy that is practically 100%. However, in the text, the accuracy is said to be 93.78 %. This is a major concern.

Does Fig 5g include the effects of variability of the neuron devices? I understand that it does not, and that would be a problem, as this variability might have worse effects than the one of synapses.

The caption of Figure 5 lacks details.

The methods section should include the methods associated with Figure 5.

Reviewer #1 (Remarks to the Author):

This paper describes the use of a previously demonstrated MoS₂ nonvolatile transistor for use in neural network processing circuits. The paper implements a Bayesian Neural Net using MoS₂ neurons and synapses. The BNN is used to explore uncertainty quantification. The accuracy is assessed using LTSpice on the IRIS dataset and with certain configurations is found to be reasonably high.

The topic of novel devices to implement BNNs is of interest to the community, as is the quantification of uncertainty in neural inference. This paper is very well written and well organized. However, before considering for publication in Nature Communications, there are some key issues which should be considered to clarify the novelty and significance of the results.

We are happy to know that the reviewer finds our manuscript is well written and is of interest to the community. A point-by-point response to the comments raised by the reviewer can be found below.

1. The authors spend a significant amount of time comparing the devices to two terminal memristors, and listing the advantages in that respect. However, the MoS₂ transistor as described is a three terminal charge trapping memory, and hence not particularly different from a standard three terminal charge trapping memory. While it is impressive that the authors have developed a CTM devices that incorporates MoS₂, it appears the circuits would all be possible to implement in a standard, commercial CTM memory, such as TANOS, SONOS, etc. The random distributions also seem likely an effect of charge trapping, which would be possible in a standard CTM. Please elaborate on whether this is the case, or there is something significantly different about the physics of the MoS₂ that cannot be replicated in a standard CTM memory.

We would like to thank the reviewer for raising this point. We agree that our Bayesian neural network (BNN) architecture can be implemented with standard three-terminal charge-trap flash memories such as TaN-Al₂O₃-Si₃N₄-SiO₂-Si (TANOS), Si-SiO₂-Si₃N₄-SiO₂-Si (SONOS) [1-3]. The random distributions are observed as an effect of random nature of charge trapping which is

typically observed in charge-trap memory (CTM) devices [4, 5]. Hence, our implementation can be easily translated to other CTM devices. Our MoS₂ memtransistors offer an alternative to other CTM, such as TANOS and SONOS. The atomically thin two-dimensional (2D) semiconductors allow geometric miniaturization of field-effect transistors (FETs) without any loss of electrostatic integrity. The scalability of FETs is captured through the screening length (λ_{SC}), shown in Eq. R1, which represents the competition between gate and drain potential for control of the channel charge.

$$\lambda_{SC} = \sqrt{\frac{\epsilon_s}{\epsilon_{ox}} t_s t_{ox}} \quad [R1]$$

Here, t_s and t_{ox} are the thicknesses and ϵ_s and ϵ_{ox} are the dielectric constants of the semiconducting channel and the insulating oxide, respectively. To avoid short channel effects the channel length of an FET (L_{CH}) has to be at least three times higher than the screening length, i.e. $L_{CH} > 3\lambda_{SC}$ [6]. The atomically thin semiconducting monolayers allows extreme scalability, as they offer t_s lower than 1 nm. Among various semiconducting 2D materials, MoS₂ has gained the most attention owing to its dominant n-type transport, stability, and ease of high-quality growth [7-10]. Hence, we have utilized MoS₂ memtransistors for the implementation of BNN.

We have clarified that the circuits can also be implemented with standard flash memories in the main manuscript.

2. The "tanh circuit" created by using T1 as a resistive load is a common drain amplifier, which is a standard transistor amplifier circuit with a well-known transfer function. I would recommend acknowledging this because it seems a bit like this is being presented as a novel circuit (which would seem odd to the microelectronic circuits community). I have also seen similar common drain circuit implemented with floating gate devices, giving similar VTC behavior, such as in Fig 5 of the referenced paper below.

We agree with the reviewer that this is a common-drain amplifier circuit. While the output characteristics of the common-drain amplifier circuit is well-known [11], to the best of our knowledge, it has not been used in the context of neuromorphic computing in order to implement the tanh/sigmoid activation function. As demonstrated in the manuscript, this simple and energy

efficient circuit can be used to implement the tanh activation function which is otherwise implemented using look-up-tables (LUT) consisting of numerous transistors [3].

We have clarified that the output characteristics is well-known and we have added the reference suggested by the reviewer in the main manuscript.

3. For the LTSpice simulations, can you elaborate on the tanh function implementation; is this just modeled with a single resistor and NMOS? If the VTC looks reasonable (i.e. Fig 4h) then why is this a major source of inaccuracy? If this circuit is so inaccurate, why would it be used rather than the referenced previously used NMOS/PMOS version in ref 39?

Figure R1. a) Schematic of circuit for the modified tanh activation function using a n -type MoS_2 memtransistor (T1) and a V -doped p -type WSe_2 memtransistor (T2), where the input voltage (V_S) is applied to the gate terminal of T1 and T2. b) The transfer characteristics of the circuit (solid line) i.e., output voltage (V_O) versus V_S , closely models the tanh activation function (dotted line). c) Schematic of circuit for the modified sigmoid activation function and its d) transfer characteristics.

We thank the reviewer for raising this question. As the reviewer rightly points out, the tanh activation function is modelled using a resistor and an n -type FET, which results in an asymmetry in the transfer function. We agree that an implementation using the integration of an n -type FET and a p -type FET would result in better accuracy. Hence, we demonstrate a circuit for a modified tanh (m-tanh) activation function using a n -type MoS_2 memtransistor (T1) and a V -doped p -type WSe_2 memtransistor (T2) as shown in Fig. R1a. The transfer function of the circuit i.e., output voltage (V_O) versus input voltage (V_S) closely follows the tanh activation function as shown in Fig. R1b. The maximum of the m-tanh activation function is determined by the drain voltage (V_{DD}). Here, V_S is applied to the gate of T1 and T2. T1 and T2 are programmed to ensure that the m-tanh function passes through the origin. Note that when $V_S = -2$ V, T1 operates in the off-state and T2

operates in the on-state, resulting in $V_O = -V_{DD}$, whereas for $V_S = 2\text{ V}$, T1 operates in the on-state and becomes more conductive than T2, which results in $V_O = V_{DD}$. While this output characteristics is well-known [3, 12], to the best of our knowledge it has not been used to implement activation functions, as the activation functions are typically implemented using look-up-tables [3]. Additionally, modified sigmoid activation function can be realized by applying 0 V to the drain terminal of T1, as shown in Fig. R1c and Fig. R1d.

We have revised the implementation of the activation function and added this discussion in the main manuscript.

4. A minor, optional note on the amplifier circuits, the more common convention is to have the high voltage on the top of the circuit such that current flows down. (It is opposite of this in Fig 4(g) and Supplemental Fig 7(a)).

We thank the reviewer for this suggestion. We have changed the figures to follow this common convention.

5. How was device to device variation modeled in LTSpice? Is there an experimental link between physical nonidealities and the modeling factors should be presented? Also, do the authors know the programming error of the devices, i.e. the delta between the expected and actual conductance after programming?

We are happy to provide further clarification. The synapses consisting of 2 memtransistors, T_+ and T_- are modelled using resistors, as shown in Fig. R2a. Here, T_+ is modelled as a resistor which draws from Gaussian conductance distribution given by $G_+ \sim N(\mu_{G_+}, \sigma_{G_+})$ and T_- is modelled as a resistor set to a constant conductance of G_- . This results in the synapse having an effective conductance distribution given by $G_{eff} \sim N(\mu_{G_{eff}}, \sigma_{G_{eff}})$, where, $\mu_{G_{eff}} = \mu_{G_+} - G_-$ and $\sigma_{G_{eff}} = \sigma_{G_+}$, enabling independent control of the mean and standard deviation of G_{eff} .

Figure R2. a) Schematic of the GRNG-based synapse (left). The input to the synapse, V_{in} is applied as $+V_{in}$ and $-V_{in}$ to the memtransistors, T_+ and T_- with conductance G_+ and G_- (modulated using V_{G_+} and V_{G_-}), respectively. The effective conductance of this synapse is given by $G_{eff} = G_+ - G_-$, allowing positive and negative conductance. On the right side is its implementation in LTSpice, where both memtransistors are implemented using resistors. b) Device-to-device variation across 40 MoS₂ memtransistors for different program and erase voltages. Relationship between the change in the drain current (ΔI_{DS}) as a result of c) erase and d) program operation and the starting current ($I_{DS,start}$). e) Histogram of $\Delta I_{DS}/I_{DS,start}^q$ for the erase operation, following a Gaussian distribution.

While the cycle-to-cycle variation in programming is useful for the generation of Gaussian random numbers, the device-to-device variation is undesirable. Fig. R2b demonstrates the device-to-device variation across 40 MoS₂ memtransistors while performing program/erase operation. Here, the progressively higher programming voltage pulses (V_P) followed by progressively higher erase voltage pulses (V_E) are applied to the MoS₂ memtransistors and the drain-to-source current (I_{DS}) is measured at a read voltage (V_R) of 0 V. This is repeated for 2 cycles, to demonstrate cycle-to-cycle variation. While the cycle-to-cycle variation is beneficial for a BNN, device-to-device variation is detrimental to its operation. Fig. R2c and Fig. R2d demonstrates the relationship between the change in current (ΔI_{DS}) as a result of erase or program operation and the starting current ($I_{DS,start}$) before each erase and program operation, respectively. They follow a power law relationship given by Eq. R2.

$$\Delta I_{DS} = p * I_{DS,start}^q \quad [R2]$$

Here, p is the scaling factor and q is the exponential factor. If we account for this dependence, the device-to-device variation can be reduced. $\Delta I_{DS}/I_{DS,start}^q$ follows a Gaussian distribution, as shown representatively in Fig. R2e for the erase operation.

In order to evaluate the effect of device-to-device variation, we implement up to 10 % variation in the parameters μ_{G+} , σ_{G+} , and G_- , which results in a corresponding variation in G_{eff} . In line with the above discussion, they are drawn from Gaussian distributions where the mean is given by expected μ_{G+} , σ_{G+} , and G_- and standard deviation of up to 10 % is considered. In the revised manuscript, we have also modelled the effect of device-to-device variation in the neurons. For this, we assume up to 10 % variation in the threshold voltage for the n -type FET ($V_{TH,n}$) and the p -type FET ($V_{TH,p}$). We have also implemented up to 10 % variation in resistance of the sense resistor.

We have included this discussion in the supplementary information and main manuscript.

6. What is the gate voltage (or gate voltage range) of the synapses during read? What operating region is the transistor in at this gate voltage? It sounds like these are modulated, but in LTSpice, if you are treating the two-device pair as a resistor, I am assuming the gate to source and drain to source voltages are such that both devices are operating in the linear part of the triode region, such that a resistor is a suitable element to represent the pair. However, this is unclear from the description.

We are happy to provide further clarification. A gate-to-source voltage (V_{GS}) or read voltage (V_R) of 0 V is used during the read step. V_R of 0 is used in order to avoid read-disturb caused by a positive or negative V_{GS} . As the reviewer rightly pointed out, both the V_{GS} and drain-to-source voltage (V_{DS}) are used such that the transistors are operating in the linear part of the triode region. Hence, they are modelled using resistors.

We have added this clarification in the main manuscript.

7. The authors should model something larger than the IRIS dataset, which is quite far from a dataset and network that would be used in a real-world application. It has been observed that have seen that excellent accuracy on small datasets does not translate to high accuracy on larger datasets. There is little if any practical application for using a custom, efficient accelerator to process a dataset as small as IRIS. I would understand the very small dataset if this was an experimental demonstration, but given that it is a simulation only, the authors should include something larger.

We appreciate the suggestion by the reviewer. We have now implemented a BNN classifier to classify the PIMA Indian diabetes dataset. This dataset consists of nine parameters such as number of pregnancies, glucose levels, insulin levels, body mass index, age, etc. To classify this dataset, we use a fully connected $8 \times 10 \times 2$ BNN i.e., it has an input layer with 8 neurons, one hidden layer with 10 neurons, and an output layer with 2 neurons. The dataset with 767 instances is divided into 720 for training and 47 for testing. *Bayes by Backprop* algorithm, with a Gaussian prior is used to train the synaptic weight distributions [13, 14]. The BNN is trained off-chip for 300 epochs as shown in Fig. R3a, to obtain train accuracy of 75.41 % and test accuracy of 80.85 %. Similar accuracy numbers have been reported in prior works [15-17].

Following Eq. R3, the output of the BNN accelerator is sampled $Z=100$ times to obtain a predictive distribution, and its mean is used to make the classification.

$$p(y^*|x^*, D) \approx \frac{1}{Z} \sum_{z=1}^Z p(y^*|x^*, W^z) \quad [R3]$$

Here, x^* and y^* are the test input and output, respectively, D is the training data, and W^z represents the z^{th} Monte Carlo weight sample. The softmax of predictive distribution can be used to calculate the uncertainty in classification or entropy given by Eq. R4. [18, 19].

$$H(p(\hat{y}^*|x^*, D)) = - \sum_{j=1}^J p_j(\hat{y}^*|x^*, D) * \log(p_j(\hat{y}^*|x^*, D)) \quad [R4]$$

Here, \hat{y}^* is the softmax output and J is the number of output classes. The entropy can be decomposed into epistemic entropy i.e., uncertainty in model and aleatoric entropy i.e., uncertainty in data as shown in Eq. R5.

$$H(p(\hat{y}^*|x^*, D)) = \Pi(p(\hat{y}^*|x^*, D)) + E_{W \sim q(W; \theta)}[H(p(\hat{y}^*|x^*, W))] \quad [R5]$$

Figure R3. a) Training and testing curves for 300 epochs of the BNN constructed to classify the PIMA Indians diabetes dataset. b) Accuracy and predicative accuracy as function of model variation. Here, the effect of variation in synaptic devices, sense resistors, and activation function is demonstrated. c) Total entropy, aleatoric entropy and epistemic entropy as a function of model variation. d) Accuracy and predicative accuracy as a function of input variation. e) Total entropy, aleatoric entropy and epistemic entropy as a function of input variation.

Here, on the right-hand side, the first term represents epistemic entropy and the second term represents aleatoric entropy. Aleatoric entropy is the average entropy for fixed weights and hence the uncertainty arises from the data. Epistemic entropy can be obtained by subtracting aleatoric entropy from total entropy, following Eq. R5.

The BNN accelerator to classify PIMA Indian diabetes dataset is evaluated using LTSpice simulations. The synapses consisting of 2 memtransistors, T_+ and T_- are modelled using resistors. Here, T_+ is modelled as a resistor which draws from Gaussian conductance distribution given by $G_+ \sim N(\mu_{G_+}, \sigma_{G_+})$ and T_- is modelled as a resistor set to a constant conductance of G_- . The sense transistor is modelled using a resistor. The tanh activation function is implemented using the combination of an n -type FET and a p -type FET. Using the BNN accelerator, we are able replicate the test accuracy of 80.85 %. In order to evaluate the effect of device-to-device variation in memtransistors, we implement up to 10 % variation for the parameters of the synapse: μ_{G_+} , σ_{G_+} , and G_- , which results in a corresponding variation in G_{eff} , the parameters of the tanh function: $V_{TH,n}$ and $V_{TH,p}$, and for the n -type FET and the p -type FET, and the conductance of the sense resistor G_S . These parameters are drawn from Gaussian distributions where the mean is given by

expected parameter value and standard deviation of up to 10 % is considered. Fig. R3b shows the effect of device-to-device (model) variation on the test accuracy and predictive accuracy. The predictive accuracy demonstrates how well the predictions of the expected correct classes are made. Here, the BNN is simulated and averaged over 5 runs. While, we observe a decrease in the test accuracy, it is not seen to significantly impact the operation of the BNN and an accuracy of $\approx 60\%$ is maintained for 10 % variation. In a BNN, we can use entropy estimation and entropy decomposition to quantify uncertainty and to find its source, respectively. Fig. R3c shows the total entropy, aleatoric entropy, and epistemic entropy as a function of model variation, calculated using Eq. R4 and Eq. R5. Here, we would expect the aleatoric uncertainty to remain unchanged and total entropy to increase. However, their extraction is impacted by the increased model variation. Nevertheless, as expected the epistemic entropy increases as the model variation increases. As shown in Fig. R3d and Fig. R3e, an increase in the input variation results in the degradation of accuracy, along with an increase in the total and aleatoric entropy, while a constant epistemic entropy is maintained, as expected. Hence, in addition to the estimation of total entropy using a BNN, with uncertainty decomposition various sources of entropy can be identified.

We have changed the classification dataset and added this discussion in the main manuscript.

8. Although the title conveys an interesting prospect, it does not seem well connected to the content and I would suggest modifying it. The work aims to quantify uncertainty, but does not "avoid inference inaccuracy".

In line with the reviewer's suggestion, we have changed the title to "A Bayesian Neural Network to Quantify Inference Inaccuracy".

9. Some of the figure captions are far too small to read. The labels on Fig 5d are equivalent to about a 4pt font.

We have ensured that the figure labels are larger and easier to read.

Reviewer #2 (Remarks to the Author):

In this work, Sebastian and Das introduce a MoS₂-based device that can provide synaptic and neuronal functions to implement hardware Bayesian neural networks. These uses are illustrated in a simulation of a crossbar circuit. The device in itself is similar to previous works by the same group, but it is used in a new manner. The idea of exploiting stochastic effect inherent to device physics to implement probabilistic AI, as is done here, is a very strong concept, which has started to develop in recent literature. The device introduced by the authors has some nice features. However, I have concerns about the manuscript and some questions about the meaning of the work.

We are happy to know that the reviewer appreciates the idea of exploiting stochasticity in devices to implement probabilistic computing demonstrated in the paper. A point-by-point response to the comments raised by the reviewer can be found below.

1. On a general note, I actually do not understand the benefits of using 2-D materials in this project. The authors explain: “The choice of MoS₂ as the element of memristor is motivated by recent demonstrations highlighting the technological viability of 2D materials [27-29] and their wide scale adoption in brain-inspired computing [30-34].” This is too vague, in my opinion. In fact, the adoption of 2-D materials in brain-inspired computing is not that wide-scale: Refs 30-34 all have S. Das as senior author, if I am not mistaken.

We are happy to add further references supporting our statement. Various memristors, transistors and other devices based on 2D materials have been explored for neuromorphic computing applications [20-22]. 2D materials have also been explored for optoelectronic synapses enabled by their optically active monolayers [23, 24]. Also, note that the atomically thin 2D semiconductors allow geometric miniaturization of FETs without any loss of electrostatic integrity. The scalability of FETs is captured through λ_{SC} , shown in Eq. R6, which represents the competition between gate and drain potential for control of the channel charge.

$$\lambda_{SC} = \sqrt{\frac{\epsilon_s}{\epsilon_{ox}} t_s t_{ox}} \quad [R6]$$

Here, t_s and t_{ox} are the thicknesses and ϵ_s and ϵ_{ox} are the dielectric constants of the semiconducting channel and the insulating oxide, respectively. To avoid short channel effects, L_{CH} has to be at least three times higher than the screening length, i.e. $L_{CH} > 3\lambda_{SC}$ [6]. The atomically thin semiconducting monolayers allows extreme scalability, as they offer t_s lower than 1 nm. Among various semiconducting 2D materials, MoS₂ has gained the most attention owing to its dominant n-type transport, stability, and ease of high-quality growth [7-10]. Hence, we have utilized MoS₂ memtransistors for the implementation of BNN.

We have added more references to support our statement in the main manuscript.

2. The title of the article, “A Neural Network Accelerator to Avoid Inference Inaccuracy”, does not seem related to the paper results. The authors do not present a neural network accelerator, and the paper never really talks about avoiding inference inaccuracy. This was a little surprising.

In line with the reviewer’s suggestion, we have changed the title to “A Bayesian Neural Network to Quantify Inference Inaccuracy”.

3. The authors should position this new work more clearly with regards to their paper "Gaussian synapses for probabilistic neural networks" (Ref 30), which has some similar keywords and ideas (but is different).

A major difference between our implementation of a probabilistic neural network (PNN) [25] and BNN is the nature of the Gaussian synapse. To implement the PNN, we design a Gaussian synapse can mimics the Gaussian function. Fig. R4a shows the schematic of the two transistor Gaussian synapse based on the integration of *n*-type MoS₂ and *p*-type BP back-gated FETs. Figure R4b also shows the equivalent circuit diagram for the Gaussian synapse, which simply consists of two variable resistors in series. The two variable resistors, i.e., R_{MoS_2} and R_{BP} correspond to the MoS₂ and BP FETs. Figure R4c shows the experimentally measured transfer characteristics i.e., I_{DS} versus V_{GS} of the Gaussian synapse for different V_{DS} . Here, the total current flowing through the series combination of MoS₂ and BP FETs is measured. Clearly, the transfer characteristics

Figure R4. a) Schematic of the GRNG-based synapse (left). The input to the synapse, V_{in} is applied as $+V_{in}$ and $-V_{in}$ to the memtransistors, T_+ and T_- with conductance G_+ and G_- (modulated using V_{G_+} and V_{G_-}), respectively. The effective conductance of this synapse is given by $G_{eff} = G_+ - G_-$, allowing positive and negative conductance. On the right side is its implementation in LTSpice, where both memtransistors are implemented using resistors. b) Device-to-device variation across 40 MoS₂ memtransistors for different program and erase voltages. Relationship between the change in the drain current (ΔI_{DS}) as a result of c) erase and d) program operation and the starting current ($I_{DS,start}$). e) Histogram of $\Delta I_{DS}/I_{DS,start}$ for the erase operation, following a Gaussian distribution.

resemble a Gaussian function which can be modeled. The emergence of Gaussian transfer characteristics can be explained using the experimentally measured transfer characteristics of its constituents, i.e., the MoS₂ FET and the BP FET, as shown in Figure R4d and Figure R4d, respectively. MoS₂ FETs exhibit unipolar n -type characteristics, whereas, BP FETs are predominantly p -type with large work function contact metals such as Ni [7, 26-28].

For the implementation of the BNN, we design synapses that can draw random numbers from a Gaussian distribution. Fig. R4e shows the design of our GRNG-based synapse with independent control over its mean (μ) and standard deviation (σ), using two MoS₂ memtransistors, T_+ and T_- . Here, the input to the synapse, V_{in} is applied as $+V_{in}$ and $-V_{in}$ to T_+ and T_- , respectively, as shown in Fig. R4e. The current at the output node, I_{out} is then given by sum of currents through T_+ and T_- i.e., I_{T_+} and I_{T_-} , according to the KCL. To control $\mu_{G_{eff}}$ and $\sigma_{G_{eff}}$, T_+ is subjected to successive erase-program-read pulse cycles, while T_- is programmed to a given state and subsequently only read, using the waveforms shown in Fig. R4f. This results in G_+ being drawn from a Gaussian distribution, with $\mu_{G_+} = 5$ nS and $\sigma_{G_+} = 0.49$ nS i.e., $G_+ \sim N$ and G_- having a constant value of \approx

8.89 nS, as shown in Fig. R4f. G_{eff} is expected to be drawn from a distribution with $\sigma_{G_{\text{eff}}} = \sigma_{G_+}$ and $\mu_{G_{\text{eff}}} = \mu_{G_+} - G_-$. This is confirmed by our measurements as shown in Fig. R4g, $G_{\text{eff}} \sim N(-3.9, 0.49)$ nS. As evident, while we have some similar keywords in both works, even the nature of synapse is very different between our PNN and BNN work. Additionally, PNNs and BNNs are both different kind of networks based on different algorithms for training such as expectation maximization and backpropagation, respectively with different network structure. In terms of the structure and operation, BNNs are closer to deep artificial neural networks, with the addition of synapses being represented by probability distributions.

4. The programming voltages are very high, similar to FLASH memory, and much higher than memristors. The authors should benchmark their device with alternative approaches (FLASH, non-2D memristors, phase change memory...).

We are happy to benchmark our devices with other alternative approaches. Table R1 shows the comparison in energy consumption between different memory technologies such as memristor, phase-change memory (PCM), NAND and our 2D memtransistors [29]. While the program/erase voltages and times for the transistor memory technologies are high, the energy consumption is similar as the programming current is much lower for the transistor technologies ($\approx 10^{-11}$ A) compared to memristors and PCM. Also note that, while our demonstration uses 2D memtransistors, it can also be implemented using commercial NAND technologies.

Table R1. Memory Technologies Energy Benchmark				
	Memristor	PCM	NAND	2D Memtransistor
Cell Elements	1T1R	1T1R	1T	1T
Read Time (ns)	<50	<60	<50	<50
Read Voltage (V)	<3	3	2	0
Program/Erase Time (ns)	<250	60	10^6	10^5
Program/Erase Voltage (V)	<3	3	15	13
Program/Erase Energy (fJ)	<50	6×10^3	10	10

We have included this table in the supplementary information.

5. I have concerns about the proposed crossbar architecture. I understand that for each presented input, the crossbar devices need to be reprogrammed multiple times to provide a distribution at the output (once per sample). Having to reprogram the crossbar numerous times to perform a single inference seems an enormous energy cost. Due to this concern, the paper should include an energy analysis with some benchmarks, in my opinion.

We thank the reviewer for their recommendation. We have now included an energy analysis. The total energy consumption of the BNN accelerator is given by Eq. R7.

$$E_{\text{Total}} = E_{\text{syn}} + E_{\text{sense}} + E_{\text{tanh}} + E_{\text{P}} + E_{\text{E}} \quad [\text{R7}]$$

$$= \sum_N [(I_{\text{T}+} + I_{\text{T}-})V_{\text{in}}t]_{\text{syn}} + [I_{\text{S}}V_{\text{S}}t]_{\text{sense}} + [(I_{\text{T}1} + I_{\text{T}2})V_{\text{DD}}t]_{\text{tanh}} + E_{\text{P}} + E_{\text{E}} \quad [\text{R7}]$$

$$E_{\text{P}} + E_{\text{E}} = \sum_{N-100} t(I_{\text{P}}V_{\text{P}} + I_{\text{E}}V_{\text{E}}) + \sum_{100} [Qt(I_{\text{P}}V_{\text{P}} + I_{\text{E}}V_{\text{E}})]_{\text{P,E}} \quad [\text{R7}]$$

Here, E_{Total} represents the total energy consumption per test sample. E_{syn} , E_{sense} , and E_{tanh} represents the total energy consumption by the synapses, sense resistors, and the tanh circuit, respectively, during the inference step. I_{S} represents the current through the sense resistor and N is the total number of devices. E_{P} and E_{E} represents the energy consumption for programming and erasing operation on these components. I_{P} and I_{E} respectively, are the gate currents associated with programming and erase operations. Here, only half of the synaptic devices (100 devices) needs to be programmed and erased for $Z=100$ Monte Carlo samples. The rest of the synaptic devices, sense transistors, and tanh circuit components are programmed and erased once and subsequently only used for inference. For E_{P} and E_{E} evaluations, maximum I_{P} , V_{P} , I_{E} , V_{E} values expected are used. Using Eq. R7, E_{syn} of 18 nJ, E_{sense} of 0.025 nJ, and E_{tanh} of 0.012 nJ are obtained from LTSpice simulations and $E_{\text{P}} + E_{\text{E}}$ of 0.34 nJ is estimated for a total of $Z=100$ Monte Carlo samples, resulting in an E_{Total} of 18.37 nJ. Note that while the program/erase voltages are high the energy consumption associated with them is low as the gate currents are on the order of pA and since the transistors are biased with a drain voltage of 0 V during programming/erasing, there is no drain current.

We have included the energy analysis main manuscript.

6. These multiple programming operations also raise the issue of device endurance, which should be discussed.

The reviewer raises an important point. For the use of MoS₂ memtransistors for BNN, it's important to evaluate its endurance. Fig. R5 shows the endurance characteristics of a MoS₂ memtransistor for a total of 2000 cycles. Here, for each cycle the device is switched between a high current state and a low current state by using a program voltage (V_P) of -11 V and erase voltage (V_E) of 11 V. The current values are read for V_R of 0 V. Minimal degradation is observed after 2000 cycles and the distinction between the high and low states are maintained.

We have included the endurance characteristics in the main manuscript.

Figure R5. Endurance characteristics of an MoS₂ memtransistor for 2000 cycles.

7. Also, if the synapses need to be reprogrammed each sample, we need additional memory arrays storing the mean value and standard deviations of each weight. This is an important cost, and this would also limit the energy efficiency of the authors' approach, as important data movement will be involved.

We are happy to provide further clarification. We propose to use a scheme where all devices are erased with a high V_E of 13 V. Following that, each device must be programmed with the separate

V_P values to set them to their expected states. Hence, we agree that we will need to store the V_P values to obtain the required mean and standard deviation.

We have included this clarification in the main manuscript.

8. A significant concern is that Fig 5 shows no Bayesian result: we only see the final accuracy, but no distributions.

Figure R6. a) Training and testing curves for 300 epochs of the BNN constructed to classify the PIMA Indians diabetes dataset. b) Accuracy and predicative accuracy as function of model variation. Here, the effect of variation in synaptic devices, sense resistors, and activation function is demonstrated. c) Total entropy, aleatoric entropy and epistemic entropy as a function of model variation. d) Accuracy and predicative accuracy as a function of input variation. e) Total entropy, aleatoric entropy and epistemic entropy as a function of input variation.

We are happy to include more Bayesian results. We have now implemented a BNN classifier to classify the PIMA Indian diabetes dataset. This dataset consists of nine parameters such as number of pregnancies, glucose levels, insulin levels, body mass index, and age. To classify this dataset, we use a fully connected $8 \times 10 \times 2$ BNN i.e., it has an input layer with 8 neurons, one hidden layer with 10 neurons, and an output layer with 2 neurons. The dataset with 767 instances is divided into 720 for training and 47 for testing. *Bayes by Backprop* algorithm, with a Gaussian prior is used to train the synaptic weight distributions [13, 14]. The BNN is trained off-chip for 300 epochs as

shown in Fig. R6a, to obtain train accuracy of 75.41 % and test accuracy of 80.85 %. Similar accuracy numbers have been reported in prior works [15-17].

Following Eq. R8, the output of the BNN accelerator is sampled $Z=100$ times to obtain a predictive distribution, and its mean is used to make the classification.

$$p(y^*|x^*, D) \approx \frac{1}{Z} \sum_{z=1}^Z p(y^*|x^*, W^z) \quad [R8]$$

Here, x^* and y^* are the test input and output, respectively, D is the training data, and W^z represents the z^{th} Monte Carlo weight sample. The softmax of predictive distribution can be used to calculate the uncertainty in classification or entropy given by Eq. R9. [18, 19].

$$H(p(\hat{y}^*|x^*, D)) = - \sum_{j=1}^J p_j(\hat{y}^*|x^*, D) * \log(p_j(\hat{y}^*|x^*, D)) \quad [R9]$$

Here, \hat{y}^* is the softmax output and J is the number of output classes. The entropy can be decomposed into epistemic entropy i.e., uncertainty in model and aleatoric entropy i.e., uncertainty in data as shown in Eq. R10.

$$H(p(\hat{y}^*|x^*, D)) = \Pi(p(\hat{y}^*|x^*, D)) + E_{W \sim q(W; \theta)}[H(p(\hat{y}^*|x^*, W))] \quad [R10]$$

Here, on the right-hand side, the first term represents epistemic entropy and the second term represents aleatoric entropy. Aleatoric entropy is the average entropy for fixed weights and hence the uncertainty arises from the data. Epistemic entropy can be obtained by subtracting aleatoric entropy from total entropy, following Eq. R10.

The BNN accelerator to classify PIMA Indian diabetes dataset is evaluated using LTSpice simulations. The synapses consisting of 2 memtransistors, T_+ and T_- are modelled using resistors. Here, T_+ is modelled as a resistor which draws from Gaussian conductance distribution given by $G_+ \sim N(\mu_{G_+}, \sigma_{G_+})$ and T_- is modelled as a resistor set to a constant conductance of G_- . The sense transistor is modelled using a sense resistor. The neuron is implemented using the combination of an n -type FET and a p -type FET. Using the BNN accelerator, we are able replicate the test accuracy of 80.85 %. In order to evaluate the effect of device-to-device variation in memtransistors, we implement up to 10 % variation for the parameters of the synapse: μ_{G_+} , σ_{G_+} , and G_- , which results in a corresponding variation in G_{eff} , the parameters of the neuron: $V_{TH,n}$ and $V_{TH,p}$, and for the n -type FET and the p -type FET, and the conductance of the sense resistor G_S . These parameters are drawn from Gaussian distributions where the mean is given by expected parameter value and

standard deviation of up to 10 % is considered. Fig. R6b shows the effect of device-to-device (model) variation on the test accuracy and predictive accuracy. The predictive accuracy demonstrates how well the predictions of the expected correct classes are made. Here, the BNN is simulated and averaged over 5 runs. While, we observe a decrease in the test accuracy, it is not seen to significantly impact the operation of the BNN and an accuracy of $\approx 60\%$ is maintained for 10 % variation. In a BNN, we can use entropy estimation and entropy decomposition to quantify uncertainty and to find its source, respectively. Fig. R6c shows the total entropy, aleatoric entropy, and epistemic entropy as a function of model variation, calculated using Eq. R4 and Eq. R5. Here, we would expect the aleatoric uncertainty to remain unchanged and total entropy to increase. However, their extraction is impacted by the increased model variation. Nevertheless, as expected the epistemic entropy increases as the model variation increases. As shown in Fig. R6d and Fig. R6e, an increase in the input variation results in the degradation of accuracy, along with an increase in the total and aleatoric entropy, while a constant epistemic entropy is maintained, as expected. Hence, in addition to the estimation of total entropy using a BNN, with uncertainty decomposition various sources of entropy/uncertainty can be identified.

We have added this discussion in the main manuscript.

9. Also, why are Bayesian neural networks useful for this example? I think that conventional neural networks get excellent accuracy on this task.

We thank the reviewer for their suggestion. As discussed in detail for the previous question, we have now implemented BNN classification on the PIMA Indian diabetes dataset, where we have a test accuracy of 80.85 %. This dataset has typically demonstrated similar accuracy numbers as evident from prior works [15-17], and hence serves better to demonstrate uncertainty estimation.

We have now implemented BNN to classify the PIMA Indian dataset and revised the main manuscript.

10. The degradation in accuracy between software and LTSpice simulation is quite severe, even for a very simple task, due to the neuron behavior. This is an important limitation. Can this problem be fixed?

Figure R7. a) Schematic of circuit for the modified tanh activation function using a n -type MoS₂ memtransistor (T1) and a V -doped p -type WSe₂ memtransistor (T2), where the input voltage (V_S) is applied to the gate terminal of T1 and T2. b) The transfer characteristics of the circuit (solid line) i.e., output voltage (V_O) versus V_S , closely models the tanh activation function (dotted line). c) Schematic of circuit for the modified sigmoid activation function and its b) transfer characteristics.

We thank the reviewer for raising this question. As the reviewer rightly points out, when the tanh activation function is modelled using a resistor and an n -type FET, there is an asymmetric transfer function, which leads to a degradation in the accuracy. To avoid this, we have now implemented the tanh activation function using an n -type FET and a p -type FET, which enables us to replicate the test accuracy of 80.85 % in LTSpice simulations. For the tanh activation function, we demonstrate a circuit for a modified tanh (m-tanh) activation function using a n -type MoS₂ memtransistors (T1) and a V -doped p -type WSe₂ memtransistor (T2) as shown in Fig. R7a. The transfer function of the circuit i.e., output voltage (V_O) versus input voltage (V_S) closely follows the tanh activation function as shown in Fig. R7b. The maximum of the m-tanh activation function is determined by the drain voltage (V_{DD}). Here, V_S is applied to the gate of T1 and T2. T1 and T2 are programmed to ensure that the m-tanh function passes through the origin. Note that when $V_S = -2 \text{ V}$, T1 operates in the off-state and T2 operates in the on-state, resulting in $V_O = -V_{DD}$, whereas for $V_S = 2 \text{ V}$, T1 operates in the on-state and becomes more conductive than T2, which results in $V_O = V_{DD}$. While this output characteristics is well-known [3, 12], to the best of our knowledge it has not been used to implement activation functions, as the activation functions are typically implemented using look-up-tables [3]. Additionally, modified sigmoid activation function can be realized by applying 0 V to the drain terminal of T1, as shown in Fig. R7c and Fig. R7d.

We have revised the implementation of the activation function and added this discussion in the main manuscript.

11. Fig 5g with 0% variation shows a test accuracy that is practically 100%. However, in the text, the accuracy is said to be 93.78 %. This is a major concern.

We are happy to provide further clarification. Here, the degradation of accuracy was a result of the asymmetry in the tanh activation function implemented using an n -type FET and a resistor. By implementing a neuron with a symmetric tanh activation function, using the integration of an n -type FET and a p -type FET enabled us to replicate the test accuracy of 80.85 % in LTSpice simulations.

12. Does Fig 5g include the effects of variability of the neuron devices? I understand that it does not, and that would be a problem, as this variability might have worse effects than the one of synapses.

The reviewer raises an excellent point. Hence, we have now included the effect of neuron variation as well. In order to evaluate the effect of device-to-device variation in memtransistors, we implement up to 10 % variation for the parameters of the synapse: μ_{G_+} , σ_{G_+} , and G_- , which results in a corresponding variation in G_{eff} , the parameters of the neuron: $V_{TH,n}$ and $V_{TH,p}$, and for the n -type FET and the p -type FET, and the conductance of the sense resistor G_S . These parameters are drawn from Gaussian distributions where the mean is given by expected parameter value and

Figure R8. a) Accuracy and predicative accuracy as function of synaptic variation. Here, the effect of variation in synaptic devices and sense resistors is demonstrated. b) Accuracy and predicative accuracy as function of total model variation. Here, the effect of variation in synaptic devices, sense resistors, and activation function is demonstrated. As evident, the inclusion of activation function does not significantly impact the accuracy.

standard deviation of up to 10 % is considered. To evaluate the added impact of neuron variation, Fig. R8a shows the effect of device-to-device variation when only the weights and sense resistors are considered and Fig. R8b shows the effect of device-to-device variation when weights, sense resistors and neurons are considered. Here, to obtain each data point, the BNN is simulated and averaged over 5 runs. We can clearly see that the variation in neurons does not significantly impact the accuracy.

In error analysis, we have now included the effect neuron variation as well.

13. The caption of Figure 5 lacks details. The methods section should include the methods associated with Figure 5.

We have now included an extensive discussion in the Methods section to provide further details on Fig. 5.

References

- [1] J. K. Han, J. Oh, G. J. Yun, D. Yoo, M. S. Kim, J. M. Yu, *et al.*, "Cointegration of single-transistor neurons and synapses by nanoscale CMOS fabrication for highly scalable neuromorphic hardware," *Sci Adv*, vol. 7, Aug 2021.
- [2] S. Hwang, J. Yu, G. H. Lee, M. S. Song, J. Chang, K. K. Min, *et al.*, "Capacitor-Based Synaptic Devices for Hardware Spiking Neural Networks," *IEEE Electron Device Letters*, vol. 43, pp. 549-552, 2022.
- [3] T. P. Xiao, C. H. Bennett, B. Feinberg, S. Agarwal, and M. J. Marinella, "Analog architectures for neural network acceleration based on non-volatile memory," *Applied Physics Reviews*, vol. 7, p. 031301, 2020.
- [4] L. Danial, V. Gupta, E. Pikhay, Y. Roizin, and S. Kvatinsky, "Modeling a Floating-Gate Memristive Device for Computer Aided Design of Neuromorphic Computing," pp. 472-477, 2020.
- [5] C. Monzio Compagnoni, R. Gusmeroli, A. S. Spinelli, and A. Visconti, "Analytical Model for the Electron-Injection Statistics During Programming of Nanoscale nand Flash Memories," *IEEE Transactions on Electron Devices*, vol. 55, pp. 3192-3199, 2008.
- [6] D. J. Frank, Y. Taur, and H.-S. P. Wong, "Generalized scale length for two-dimensional effects in MOSFETs," *Electron Device Letters, IEEE*, vol. 19, pp. 385-387, 1998.
- [7] S. Das, H. Y. Chen, A. V. Penumatcha, and J. Appenzeller, "High performance multilayer MoS2 transistors with scandium contacts," *Nano Lett*, vol. 13, pp. 100-5, Jan 9 2013.
- [8] P. C. Shen, C. Su, Y. Lin, A. S. Chou, C. C. Cheng, J. H. Park, *et al.*, "Ultralow contact resistance between semimetal and monolayer semiconductors," *Nature*, vol. 593, pp. 211-217, May 2021.
- [9] (accessed Sep 1, 2020). *2D Semiconductor Transistor Trends*. Available: <http://2d.stanford.edu>
- [10] S. Y. Kim, J. Kwak, C. V. Ciobanu, and S. Y. Kwon, "Recent Developments in Controlled Vapor-Phase Growth of 2D Group 6 Transition Metal Dichalcogenides," *Adv Mater*, vol. 31, p. e1804939, May 2019.
- [11] S. Shah, H. Toreyin, J. Hasler, and A. Natarajan, "Models and Techniques for Temperature Robust Systems on a Reconfigurable Platform," *Journal of Low Power Electronics and Applications*, vol. 7, p. 21, 2017.

- [12] A. Sebastian, S. Das, and S. Das, "An Annealing Accelerator for Ising Spin Systems Based on In-Memory Complementary 2D FETs," *Adv Mater*, vol. 34, p. e2107076, Jan 2022.
- [13] J. Antoran, "Bayesian-Neural-Networks," 2019.
- [14] C. Blundell, J. Cornebise, K. Kavukcuoglu, and D. Wierstra, "Weight Uncertainty in Neural Networks," *ArXiv*, vol. abs/1505.05424, 2015.
- [15] Q. Zou, K. Qu, Y. Luo, D. Yin, Y. Ju, and H. Tang, "Predicting Diabetes Mellitus With Machine Learning Techniques," *Front Genet*, vol. 9, p. 515, 2018.
- [16] R. Vaishali, R. Sasikala, S. Ramasubbareddy, S. Remya, and S. Nalluri, "Genetic algorithm based feature selection and MOE Fuzzy classification algorithm on Pima Indians Diabetes dataset," pp. 1-5, 2017.
- [17] R. Z. Islamic, "Diagnosis of Diabetes in Female Population of Pima Indian Heritage with Ensemble of BP Neural Network and SVM," 2012.
- [18] Y. Kwon, J.-H. Won, B. J. Kim, and M. C. Paik, "Uncertainty quantification using Bayesian neural networks in classification: Application to biomedical image segmentation," *Computational Statistics & Data Analysis*, vol. 142, p. 106816, 2020.
- [19] A. Kendall and Y. Gal, "What uncertainties do we need in Bayesian deep learning for computer vision?," presented at the Proceedings of the 31st International Conference on Neural Information Processing Systems, Long Beach, California, USA, 2017.
- [20] T. J. Ko, H. Li, S. A. Mofid, C. Yoo, E. Okogbue, S. S. Han, *et al.*, "Two-Dimensional Near-Atom-Thickness Materials for Emerging Neuromorphic Devices and Applications," *iScience*, vol. 23, p. 101676, Nov 20 2020.
- [21] K. C. Kwon, J. H. Baek, K. Hong, S. Y. Kim, and H. W. Jang, "Memristive Devices Based on Two-Dimensional Transition Metal Chalcogenides for Neuromorphic Computing," *Nanomicro Lett*, vol. 14, p. 58, Feb 5 2022.
- [22] J. Bian, Z. Cao, and P. Zhou, "Neuromorphic computing: Devices, hardware, and system application facilitated by two-dimensional materials," *Applied Physics Reviews*, vol. 8, p. 041313, 2021.
- [23] L. Mennel, J. Symonowicz, S. Wachter, D. K. Polyushkin, A. J. Molina-Mendoza, and T. Mueller, "Ultrafast machine vision with 2D material neural network image sensors," *Nature*, vol. 579, pp. 62-66, Mar 2020.

- [24] P. Wu, T. He, H. Zhu, Y. Wang, Q. Li, Z. Wang, *et al.*, "Next-generation machine vision systems incorporating two-dimensional materials: Progress and perspectives," *InfoMat*, vol. 4, 2021.
- [25] A. Sebastian, A. Pannone, S. Subbulakshmi Radhakrishnan, and S. Das, "Gaussian synapses for probabilistic neural networks," *Nat Commun*, vol. 10, p. 4199, Sep 13 2019.
- [26] S. Das, M. Demarteau, and A. Roelofs, "Ambipolar phosphorene field effect transistor," *ACS nano*, vol. 8, pp. 11730-11738, 2014.
- [27] S. Das, J. A. Robinson, M. Dubey, H. Terrones, and M. Terrones, "Beyond Graphene: Progress in Novel Two-Dimensional Materials and van der Waals Solids," *Annual Review of Materials Research, Vol 45*, vol. 45, pp. 1-27, 2015.
- [28] D. S. Schulman, A. J. Arnold, and S. Das, "Contact engineering for 2D materials and devices," *Chem Soc Rev*, Mar 2 2018.
- [29] O. Kavehei, "Memristive devices and circuits for computing, memory, and neuromorphic applications," 2012.

REVIEWER COMMENTS

Reviewer #1 (Remarks to the Author):

I have reviewed the response and new manuscript, including the new tanh circuit. This version has addressed many of my previous concerns in their detailed response. However, I still would consider the following two issues with the manuscript, in determining if it is impactful enough to publish in Nature Communications.

The most significant issue with the paper is that there does not seem to be an advantage to using a MoS₂ charge trapping device for this work. The same body of work could have been demonstrated using measured data from a SONOS, TANOS, or other modern, commercial device (perhaps a floating gate cell) – almost certainly with more consistent and stable electrical behavior due to maturity (including consistent random noise). I agree with the standard scaling argument for MoS₂ transistors (as logic devices) that the authors presented in their response, but this scaling argument is not really relevant to how the device is being used in the paper. Hence, this MoS₂ is not needed to achieve the functionality demonstrated in the paper. My general impression is that the paper is implying that this novel (but previously demonstrated) MoS₂ device is enabling this new functionality – the paper must be clear that a new device is -not- required to achieve the presented functionality. Hence, it also follows that the novelty of the paper must be judged on the new circuit and application concepts, and whether these results constitute publication in Nature Comm.

In general, I did find the circuit and relevant mathematical analysis presented in Fig 5 and the relevant equations compelling and well thought out, albeit an unusual circuit. I would comment that one significant challenge of this configuration will be that it is very dependent on the programming accuracy of the resistance of the three devices at the end/bottom of each column, and programming error of these “neurons” will lead to significantly greater accuracy degradation than the same error would cause in the synapses. Put slightly differently, any noise or variation in those column end devices will essentially have equal footing to the sum of all the noise from synapses in the column.

Reviewer #2 (Remarks to the Author):

The authors have very significantly improved the quality of the manuscript. I still have a few comments.

- The endurance of 2000 is in fact low, as each inference will consume many of these cycles. Also, the endurance experiment only validates that the ON and OFF states can be distinguished, not the fact the probabilistic synapse behavior is retained. This experiment is important, I think. The endurance issue should also be clearly highlighted as one of the major limitations of the approach.

- The title of the paper is still not adapted. All Bayesian neural networks quantify accuracy, and this particular aspect is not very developed in the manuscript. A more relevant title could be something like "Two-dimensional materials-based synapses for Bayesian neural network".

- The comparison with Ref 30 that the authors gave me in their answer should be included in the manuscript.

Reviewer #1 (Remarks to the Author):

I have reviewed the response and new manuscript, including the new tanh circuit. This version has addressed many of my previous concerns in their detailed response. However, I still would consider the following two issues with the manuscript, in determining if it is impactful enough to publish in Nature Communications.

We are happy to know that we have addressed your previous concerns. A point-by-point response to the comments raised by the reviewer can be found below.

1. The most significant issue with the paper is that there does not seem to be an advantage to using a MoS₂ charge trapping device for this work. The same body of work could have been demonstrated using measured data from a SONOS, TANOS, or other modern, commercial device (perhaps a floating gate cell) – almost certainly with more consistent and stable electrical behavior due to maturity (including consistent random noise). I agree with the standard scaling argument for MoS₂ transistors (as logic devices) that the authors presented in their response, but this scaling argument is not really relevant to how the device is being used in the paper. Hence, this MoS₂ is not needed to achieve the functionality demonstrated in the paper. My general impression is that the paper is implying that this novel (but previously demonstrated) MoS₂ device is enabling this new functionality – the paper must be clear that a new device is -not- required to achieve the presented functionality. Hence, it also follows that the novelty of the paper must be judged on the new circuit and application concepts, and whether these results constitute publication in Nature Comm.

We are happy to provide further clarification. We included the following statements in the main manuscript to clarify that MoS₂ memristors can be replaced with devices with charge-trap memory.

“Here, the random distributions are observed as an effect of random nature of charge trapping which is typically observed in charge-trap memory devices [1, 2]. Hence, these memtransistors can also be replaced with standard three-terminal charge-trap flash memories such as TaN-Al₂O₃-Si₃N₄-SiO₂-Si (TANOS) and Si-SiO₂-Si₃N₄-SiO₂-Si (SONOS) [3-5].” Note that MoS₂ is being used here as it has emerged as a potential alternative to Si in recent years. Hence, MoS₂ and other 2D materials have been used as channel material for devices with floating-gate to demonstrate

non-volatile memory [6-9]. Hence, we have utilized MoS₂ memtransistors for the implementation of BNN.

2. In general, I did find the circuit and relevant mathematical analysis presented in Fig 5 and the relevant equations compelling and well thought out, albeit an unusual circuit. I would comment that one significant challenge of this configuration will be that it is very dependent on the programming accuracy of the resistance of the three devices at the end/bottom of each column, and programming error of these “neurons” will lead to significantly greater accuracy degradation than the same error would cause in the synapses. Put slightly differently, any noise or variation in those column end devices will essentially have equal footing to the sum of all the noise from synapses in the column.

We are happy to know that the reviewer finds the circuit and mathematical analysis compelling and well thought out. We agree that evaluating the programming error introduced by the neuron must be analyzed. In order to evaluate the effect of device-to-device variation in memtransistors, we implement 10 % variation for the parameters of the synapse: μ_{G_+} , σ_{G_+} , and G_- , which results in a corresponding variation in G_{eff} , the parameters of the neuron: $V_{TH,n}$ and $V_{TH,p}$, and for the n -type FET and the p -type FET, and the conductance of the sense resistor G_S . These parameters are

drawn from Gaussian distributions where the mean is given by expected parameter value and standard deviation of 10 % is considered. We separately evaluate the effect of variation in synapses and neurons on the classification

Figure R1. a) Accuracy and predicative accuracy as function of variation in synapse. b) Accuracy and predicative accuracy as function of variation in neuron. The variation in neuron includes variation in the circuit for activation function and the sense resistor.

accuracy. As evident from Fig. R1, variation in synapse is much more detrimental to the performance of the BNN compared to the variation in neuron.

In the revised manuscript, we have clarified that variation in synapses is more detrimental to the operation of the BNN.

Reviewer #2 (Remarks to the Author):

The authors have very significantly improved the quality of the manuscript. I still have a few comments.

We are happy to know that the reviewer finds that the manuscript has improved. A point-by-point response to the comments raised by the reviewer can be found below.

1. The endurance of 2000 is in fact low, as each inference will consume many of these cycles. Also, the endurance experiment only validates that the ON and OFF states can be distinguished, not the fact the probabilistic synapse behavior is retained. This experiment is important, I think. The endurance issue should also be clearly highlighted as one of the major limitations of the approach.

The reviewer raises an important point regarding low endurance and retention of synaptic behavior. For the use of MoS₂ memtransistors as a synapse for a BNN, it's important to ensure that the random number generation is maintained across multiple cycles. Hence using MoS₂ memtransistors, we demonstrate the endurance characteristics/random number generation for 20,000 cycles in **Fig. R2a**. The gate is subjected to successive erase-program-read pulses, and hence each cycle shown in **Fig. R2a** includes the effect of a program and erase operation. Here, multiple current levels consisting of high and low current states (achieved with programming voltages, V_P of -8.5 V and -8 V, respectively) for different drain-to-source voltages (V_{DS}) are demonstrated. Additionally, to demonstrate the retention of the random number generation, the moving mean (μ) and moving standard deviation (σ) for these states are demonstrated in **Fig. R2b**

Figure R2. a) Endurance characteristics of an MoS₂ memtransistor demonstrating various conductance states for a total of 20,000 cycles. b) Moving mean and c) moving standard deviation for these different states.

and **Fig. R2c**. The moving μ and σ are obtained across 100 samples at a time. Clearly the random number generation is stable, and the expected synaptic behaviors of increasing μ and σ as a function of V_{DS} and $|V_P|$ is retained after numerous measurements. However, we agree with the reviewer that the endurance must be improved for MoS₂ memtransistors and we have clarified this in the main text.

We have included this discussion in the main manuscript and supplementary information.

2. The title of the paper is still not adapted. All Bayesian neural networks quantify accuracy, and this particular aspect is not very developed in the manuscript. A more relevant title could be something like "Two-dimensional materials-based synapses for Bayesian neural network".

We thank the reviewer for their suggestion. We have now changed the tile to “Two-dimensional materials-based synapses for Bayesian neural network”.

3. The comparison with Ref 30 that the authors gave me in their answer should be included in the manuscript.

We have included this discussion in the manuscript.

References

- [1] L. Danial, V. Gupta, E. Pikhay, Y. Roizin, and S. Kvatinsky, "Modeling a Floating-Gate Memristive Device for Computer Aided Design of Neuromorphic Computing," pp. 472-477, 2020.
- [2] C. Monzio Compagnoni, R. Gusmeroli, A. S. Spinelli, and A. Visconti, "Analytical Model for the Electron-Injection Statistics During Programming of Nanoscale nand Flash Memories," *IEEE Transactions on Electron Devices*, vol. 55, pp. 3192-3199, 2008.
- [3] J. K. Han, J. Oh, G. J. Yun, D. Yoo, M. S. Kim, J. M. Yu, *et al.*, "Cointegration of single-transistor neurons and synapses by nanoscale CMOS fabrication for highly scalable neuromorphic hardware," *Sci Adv*, vol. 7, Aug 2021.
- [4] S. Hwang, J. Yu, G. H. Lee, M. S. Song, J. Chang, K. K. Min, *et al.*, "Capacitor-Based Synaptic Devices for Hardware Spiking Neural Networks," *IEEE Electron Device Letters*, vol. 43, pp. 549-552, 2022.
- [5] T. P. Xiao, C. H. Bennett, B. Feinberg, S. Agarwal, and M. J. Marinella, "Analog architectures for neural network acceleration based on non-volatile memory," *Applied Physics Reviews*, vol. 7, p. 031301, 2020.
- [6] D. Li, M. Chen, Z. Sun, P. Yu, Z. Liu, P. M. Ajayan, *et al.*, "Two-dimensional non-volatile programmable p-n junctions," *Nat Nanotechnol*, vol. 12, pp. 901-906, Sep 2017.
- [7] E. Zhang, W. Wang, C. Zhang, Y. Jin, G. Zhu, Q. Sun, *et al.*, "Tunable charge-trap memory based on few-layer MoS₂," *ACS Nano*, vol. 9, pp. 612-9, Jan 27 2015.
- [8] C. Liu, X. Yan, J. Wang, S. Ding, P. Zhou, and D. W. Zhang, "Eliminating Overerase Behavior by Designing Energy Band in High-Speed Charge-Trap Memory Based on WSe₂," *Small*, vol. 13, May 2017.
- [9] Q. Feng, F. Yan, W. Luo, and K. Wang, "Charge trap memory based on few-layer black phosphorus," *Nanoscale*, vol. 8, pp. 2686-92, Feb 7 2016.

REVIEWER COMMENTS

Reviewer #1 (Remarks to the Author):

I appreciate the authors work modeling the neural net error as a function of variation in the neuron and synapses and found that an important addition to the manuscript. However, in the model of neuron variability vs accuracy, I have found an inconsistency that I've been unable to reconcile. The finding of Fig R1b shows the sense conductance G_s (which should be specific to each column, as G_{sj}) can vary by 1000% with a negligible impact on accuracy - which is a very surprising finding and merits a detailed explanation. Mathematically, for this to be true, the G_s term in the denominator of the first summation term in eqn 4a must be negligible compared to the other term (sum) in the denominator. However, this contradicts the original purpose of G_{sj} correction - which was to correct the error associated with each column, stated as " $G_{sj}(k)$ is used to make sure that each column of the crossbar array has the same $\gamma(k)$ ". If G_{sj} , which represents the inaccuracy of each column, is negligible, then why is this compensation scheme even needed? The proposal of the compensation term implies that G_s is large enough to significantly affect the denominator of 4a. But the plot in R1b implies that G_s has a negligible affect on the accuracy. Given the strong effect of V_{thn} and V_{thp} on the output, a similar argument could be made for the very small V_t variation which was varied in fig R1b. Perhaps more details on how specifically the 10% variation was modeled would clarify this result. In any case, I strongly recommend checking and clarifying this before the article is published - specifically assert whether G_s term has a non-negligible effect on the accuracy and is needed for column compensation, or if G_s is negligible and does not need to be adjusted between columns.

Reviewer #2 (Remarks to the Author):

The authors have addressed my last comments well.

COMMENTS TO AUTHOR:**Reviewer #1 (Remarks to the Author):**

I appreciate the authors work modeling the neural net error as a function of variation in the neuron and synapses and found that an important addition to the manuscript. However, in the model of neuron variability vs accuracy, I have found an inconsistency that I've been unable to reconcile. The finding of Fig R1b shows the sense conductance G_s (which should be specific to each column, as G_{sj}) can vary by 1000% with a negligible impact on accuracy - which is a very surprising finding and merits a detailed explanation. Mathematically, for this to be true, the G_s term in the denominator of the first summation term in eqn 4a must be negligible compared to the other term (sum) in the denominator. However, this contradicts the original purpose of G_{sj} correction - which was to correct the error associated with each column, stated as " $G_{sj}(k)$ is used to make sure that each column of the crossbar array has the same $\gamma(k)$ ". If G_{sj} , which represents the inaccuracy of each column, is negligible, then why is this compensation scheme even needed? The proposal of the compensation term implies that G_s is large enough to significantly affect the denominator of 4a. But the plot in R1b implies that G_s has a negligible affect on the accuracy. Given the strong effect of V_{thn} and V_{thp} on the output, a similar argument could be made for the very small V_t variation which was varied in fig R1b. Perhaps more details on how specifically the 10% variation was modeled would clarify this result. In any case, I strongly recommend checking and clarifying this before the article is published - specifically assert whether G_s term has a non-negligible effect on the accuracy and is needed for column compensation, or if G_s is negligible and does not need to be adjusted between columns.

We are happy to know that the reviewer appreciates our response. We are happy to provide further clarification on the relevance of the sense transistor. Note that the primary purpose of the sense transistor is to perform a current-to-voltage conversion. Here, the current through a column

(corresponding to the dot-product of input and weights) is converted to a proportional voltage given by **Eq. R1**.

$$V_{Sj}^{(k)} = \frac{\sum_{i=1}^M V_i^{(k)} G_{ij}^{(k)}}{G_{Sj}^{(k)} + \sum_{i=1}^M G_{ij}^{(k)}} \quad [R1]$$

Here, $V_i^{(k)}$ is the input voltage, $G_{ij}^{(k)}$ is the conductance of synapse, and $G_{Sj}^{(k)}$ is the conductance of the sense transistor for the i^{th} row, j^{th} column and k^{th} layer. The current-to-voltage conversion allows us to seamlessly integrate the circuit for m-tanh activation function into the crossbar array, as the input to the m-tanh activation function is a voltage. The secondary purpose of the sense transistor is to ensure that the nonideality in **Eq. R1** i.e., the denominator term is consistent between different columns. Note, that for our implementation, the difference in $\sum_{i=1}^M G_{ij}^{(k)}$ between different columns is not significant and hence the $G_{Sj}^{(k)}$ correction in our case is small in magnitude. Additionally, note that each synapse is represented by two devices instead one and hence their combined conductance leads to a much higher magnitude of $\sum_{i=1}^M G_{ij}^{(k)}$ compared to $G_{Sj}^{(k)}$. In a different network implementation, where $\sum_{i=1}^M G_{ij}^{(k)}$ is significantly different between columns, the impact error in $G_{Sj}^{(k)}$ on the classification accuracy will be higher.

We believe that the impact of variation the tanh activation (implemented through variation in $V_{\text{TH},n}$ and $V_{\text{TH},p}$) is low due to the binary nature of the classification problem that we have chosen. For a multiclass classification problem, the impact of the variation in activation function for is expected to be larger. In fact, in the initial version of the manuscript, we inspected a multiclass problem, where the impact of variation in the activation function was seen to be higher. To evaluate the effect of model variation, the parameters are drawn from Gaussian distributions where the mean is given by expected parameter value and standard deviation of up to 10 % is considered.

This is achieved using gaussian random number generation in Python, with the expected parameter value and percentage of standard deviation. We have verified our implementation to ensure that the variation in the parameters is modelled correctly.

REVIEWERS' COMMENTS

Reviewer #1 (Remarks to the Author):

I thank the authors for clarifying the previous response. The authors made a clarification that seems to suggest a change in the manuscript text, but I did not see a change, and hence the statement in current version of the manuscript is not completely correct. I would recommend fixing this before publication.

Specifically, the authors have written in the rebuttal: "We believe that the impact of variation the tanh activation (implemented through variation in $VV_{TH,nn}$ and $VV_{TH,pp}$) is low due to the binary nature of the classification problem that we have chosen. For a multiclass classification problem, the impact of the variation in activation function for is expected to be larger."

This would render the statement in the manuscript incorrect: "In model variation, it was determined that the variation in synapses is more determinantal to the performance of BNN compared to variations in the neuron." From the rebuttal statement, the authors appear to believe this is not always true, but depends on the nature of the classification problem. I would recommend the authors correct this before publication.

COMMENTS TO AUTHOR:**Reviewer #1 (Remarks to the Author):**

I thank the authors for clarifying the previous response. The authors made a clarification that seems to suggest a change in the manuscript text, but I did not see a change, and hence the statement in the current version of the manuscript is not completely correct. I would recommend fixing this before publication.

Specifically, the authors have written in the rebuttal: "We believe that the impact of variation the tanh activation (implemented through variation in $VVTH,nn$ and $VVTH,pp$) is low due to the binary nature of the classification problem that we have chosen. For a multiclass classification problem, the impact of the variation in activation function for is expected to be larger."

This would render the statement in the manuscript incorrect: "In model variation, it was determined that the variation in synapses is more determinantal to the performance of BNN compared to variations in the neuron." From the rebuttal statement, the authors appear to believe this is not always true, but depends on the nature of the classification problem. I would recommend the authors correct this before publication.

We thank the reviewer for this suggestion. We have made this clarification in the main manuscript.